# Asymmetric bubble-mediated gas transfer enhances global ocean CO$_2$ uptake

Yuanxu Dong [1,2] ✉, Mingxi Yang [3], Thomas G. Bell [3], Christa A. Marandino[1] & David K. Woolf [4]

Sea-air carbon dioxide (CO$_2$) flux is typically estimated from the product of the gas transfer velocity ($K$) and the CO$_2$ fugacity difference between the ocean surface and atmosphere. Total gas exchange comprises interfacial transfer across the unbroken surface and bubble-mediated transfer from wave breaking. While interfacial transfer is symmetric for invasion and evasion, bubble-mediated transfer theoretically favours invasion due to hydrostatic pressure, though field evidence has been lacking. Here we provide direct field evidence of this asymmetry and develop an asymmetric flux equation. Applying the asymmetric equation reduces bias in $K$, and increases global oceanic CO$_2$ uptake by 0.3-0.4 Pg C yr$^{-1}$ (~15% on average from 1991 to 2020) relative to conventional estimates. Further evasion data are needed to better quantify the asymmetry factor. Our study suggests that the ocean may have absorbed more CO$_2$ than previously thought, and the asymmetric equation should be used for future CO$_2$ flux assessments.

The global ocean is a major sink of anthropogenic carbon dioxide (CO$_2$), and accurate quantification of the sea-air CO$_2$ flux is critical for projecting the future climate and developing climate mitigation strategies[1]. The exchange of CO$_2$ between sea and air is a significant component of the global carbon cycle. Sea-air CO$_2$ fluxes vary regionally and seasonally between uptake (invasion) and outgassing (evasion), leading to a net global ocean CO$_2$ uptake of ~3 Pg C yr$^{-1}$ during the last decade[2].

The sea-air exchange of sparingly soluble gases such as CO$_2$ is controlled by processes immediately below the sea surface[3]. Wind is the major forcing factor for surface turbulence in the open ocean, driving gas exchange across the sea-air interface[4]. The sea-air gas flux (*Flux*, e.g., in mol cm$^{-2}$ h$^{-1}$) is often estimated by a bulk equation:

$$Flux = K(C_w - C_a) \tag{1}$$

Total gas transfer velocity $K$ (cm hr$^{-1}$) is often normalised to a reference Schmidt number ($Sc$) to account for variability due to temperature and salinity (i.e., $K = K_{660}(Sc/660)^{-0.5}$, with $Sc$ equal to 660 at 20 °C seawater for CO$_2$) and then parametrized as a simple function of wind speed ($U_{10}$). $C_w-C_a$ (= $\Delta C$) is the gas concentration difference between the seawater ($C_w$) and the sea-air interface ($C_a$) that is equilibrated with the lower atmosphere. For CO$_2$, $\Delta C$ is often expressed as the sea-air CO$_2$ fugacity difference (i.e., $\Delta f\text{CO}_2 = f\text{CO}_{2w}-f\text{CO}_{2a}$, in µatm) multiplied by the gas solubility ($\alpha$, e.g., in mol cm$^{-3}$ µatm$^{-1}$). We refer to Eq. (1) as a "symmetric" bulk formula because the flux is proportional to $\Delta C$, regardless of the flux direction.

Wind stress leads to wave formation and development. Wave breaking entrains air into the water, creating bubbles and providing a separate pathway for gas transfer[5,6]. The total gas transfer can be mechanistically separated into interfacial transfer and bubble-mediated transfer. The interfacial transfer is symmetric for invasion and evasion fluxes, and is independent of gas solubility because it occurs at sea level air pressure with an effectively infinite air volume. In contrast, bubble-mediated transfer: (1) depends on solubility because bubbles have limited volume and lifetime[7], and (2) is asymmetric because the internal gases within submerged bubbles are over-pressured[8].

Different gases in a bubble have different characteristic equilibration times. Relatively soluble gases equilibrate faster, which limits

[1]Marine Biogeochemistry Research Division, GEOMAR Helmholtz Centre for Ocean Research Kiel, Kiel, Germany. [2]Institute of Environmental Physics, Heidelberg University, Heidelberg, Germany. [3]Plymouth Marine Laboratory, Plymouth, UK. [4]International Centre for Island Technology, Heriot-Watt University, Orkney, UK. ✉e-mail: ydong@geomar.de

the total gas transfer that can occur via bubbles. Thus, bubble-mediated transfer has a solubility dependence and is relatively more important for less soluble gases[7,9]. Furthermore, bubble-mediated transfer is more efficient for invasion than evasion[8-10]. This "asymmetric" bubble effect occurs primarily due to hydrostatic pressure. Subsurface pressure compresses a bubble, causing a concentration increase in all gases within the bubble and encouraging net transfer from the bubble into the ocean. The pressure can also drive out nitrogen and oxygen, shrinking the bubble, increasing trace gas concentrations, and encouraging additional gas transfer into the ocean. Some small bubbles may dissolve entirely, forcing the contents into the ocean completely.

Due to this asymmetric effect, an asymmetric bulk equation has been proposed for calculating the sea-air gas flux that accounts for the over-pressure effect in bubbles[8]:

$$Flux = K \left[ C_w - C_a (1 + \Delta_s) \right] \qquad (2)$$

where $\Delta_s$ is an asymmetry factor, representing the "average" fractional enhancement in the gas concentration in contact with the sea due to bubbles[11] (see "Methods" section for details). If the overall gas transfer ($K$) is mechanistically separated into the interfacial transfer component ($K_{int}$) and the bubble-mediated transfer component ($K_{bub}$)[7], Eq. (2) can be expressed as:

$$Flux = K_{int}(C_w - C_a) + K_{bub} \left[ C_w - C_a(1 + \delta) \right] \qquad (3)$$

The first term in the right side of Eq. (3) represents the interfacial transfer process, which is symmetric, whereas the second term corresponds to the bubble-mediated transfer process, which is asymmetric (represented by the over-pressure factor, $\delta$). Note that $\delta$ and $\Delta_s$ have different meanings: $\delta$ is only related to the bubble process, while $\Delta_s$ captures the combined effects of both bubble and interfacial processes. By combining Eqs. (2 and 3), $\Delta_s$ and $\delta$ can be related as:

$$\Delta_s = \delta K_{bub} / (K_{int} + K_{bub}) \qquad (4)$$

Field observations, such as the supersaturation of noble gases[12], typically reflect $\Delta_s$, since the natural measurements integrate both interfacial and bubble processes. $\delta$ can be simulated by bubble dynamic models based on the near-surface bubble size distributions[11,13].

The asymmetric transfer of highly insoluble gases, such as noble gases, is well-evidenced by observations of their supersaturation state in the field[12] and laboratory[14]. However, the saturation state cannot be used to evaluate the asymmetric transfer of $CO_2$ because of the effect of biological activity and seawater $CO_2$ buffering capacity. Previous research suggested that asymmetric bubble transfer accounts for more than 20% of the total oceanic $CO_2$ uptake based on a $CO_2$ supersaturation factor scaled from oxygen[15]. The asymmetry results of very poorly soluble gases provide an upper limit; however, extrapolating from these gases to infer asymmetric effects on $CO_2$ is likely unreliable. Alternatively, asymmetric bubble transfer can be estimated using numerical models coupled with bubble dynamic observations[11,13,16]. For sparingly soluble gases (e.g., $CO_2$), this asymmetry is mainly driven by large bubbles near the sea surface[9]. While a study argued that the asymmetric effect is insignificant for $CO_2$[8], more recent research inferred a substantial asymmetry in $CO_2$ transfer from measurements of large bubbles near the sea surface[11]. However, no results or analysis have thus far demonstrated direct evidence of asymmetric $CO_2$ transfer.

Direct flux measurements by the eddy covariance (EC) technique can be used with gas concentration observations to derive $K$ from Eqs. (1 and 2). In this study, field evidence of asymmetric bubble-mediated $CO_2$ transfer is observed in a re-analysis of a large EC dataset.

The impact of asymmetric transfer on global ocean $CO_2$ flux estimates is then assessed by comparing fluxes calculated using the symmetric bulk equation (Eq. (1)) with those calculated using the asymmetric bulk equation (Eq. (2)).

## Results

### Evidence of asymmetric $CO_2$ transfer

A large EC $CO_2$ flux and $\Delta fCO_2$ dataset (4082 h, 17 cruises, Fig. S1A) is used to evaluate asymmetric sea-air $CO_2$ transfer. The dataset contains flux observations with strong invasion ($\Delta fCO_2 \leq -20\ \mu$atm), weak invasion ($-20 < \Delta fCO_2 \leq 0\ \mu$atm), weak evasion ($0 < \Delta fCO_2 < 20\ \mu$atm), and strong evasion ($\Delta fCO_2 \geq 20\ \mu$atm). Each scenario includes data collected from multiple cruises (Fig. S1B, C). High wind speeds ($U_{10} > 12\ \text{m s}^{-1}$) were observed within all four scenarios (Fig. S2). If asymmetry has a negligible effect on $CO_2$ exchange, the transfer velocity derived from EC $CO_2$ fluxes using the symmetric bulk equation ($K_{Sy}$, Eq. (1)) should be consistent regardless of whether the $CO_2$ flux is invasive or evasive. In contrast, if the asymmetric effect is important for $CO_2$ transfer, the $CO_2$ transfer velocity computed using the symmetric bulk equation will be biased, causing $K_{Sy}$ to differ between invasion and evasion conditions, i.e., $K_{Sy}$ (weak invasion) > $K_{Sy}$ (strong invasion) > $K_{Sy}$ (strong evasion) > $K_{Sy}$ (weak evasion). From theory, this bias is expected to be largest when $\Delta fCO_2$ is small and wind speed is high (see Supplementary Information, Section 1, Eq. S4).

Traditionally, $K$ is derived by dividing the EC flux by the $\Delta C$ (i.e., $K = Flux / \Delta C$), and then parameterising $K$ against wind speed (one-dimensional (1D) fitting method). However, under weak invasion or evasion conditions (i.e., $|\Delta fCO_2| < 20\ \mu$atm), this method often fails because the large relative uncertainties in the EC flux and $\Delta fCO_2$ lead to unreliable derivations of $K$. Therefore, many authors have chosen to exclude low-$\Delta fCO_2$ data from their analysis[17] ("Methods"). However, although the relative uncertainty in EC fluxes under these conditions is large, the absolute uncertainty is small[18]. Moreover, the asymmetric effect is expected to be more pronounced under the weak invasion/evasion conditions (Eq. S4), making these data valuable. This study uses an innovative two-dimensional (2D) method to fit the $CO_2$ flux directly as a function of both wind speed and $\Delta C$, avoiding the $K$ derivation process (see "Methods"). This method enables inclusion of small-$\Delta fCO_2$ data in the parameterisation. The bulk flux derived from the 2D fitting approach generally replicates the hourly EC flux observations across various conditions (Fig. S3).

The 2D fit is first run using the symmetric bulk equation. The results show that there is a notable divergence between the parameterised $K$ ($K^{2D}_{Sy}$) for invasion and evasion conditions (Fig. 1A). These divergences agree with theory that the asymmetry is important for $CO_2$ exchange (i.e., weak invasion > strong invasion > strong evasion > weak evasion), and the discrepancies are largest at high wind speeds (Fig. 1A). Statistical analysis indicates that the discrepancies at wind speeds above 10 m s$^{-1}$ are significant ($p$-value < 0.05, Fig. S4A), except in the weak evasion case, where limited data reduce confidence in the result.

To verify whether accounting for asymmetric transfer can reconcile the difference between invasion and evasion shown in Fig. 1A, the 2D fit process is repeated using $K$ computed from the asymmetric bulk equation ($K^{2D}_{Asy}$, Eq. (2)). Before the fitting process, the asymmetric factor ($\Delta_s$) in Eq. (2) should first be determined. Here, we use two approaches to estimate $\Delta_s$: reanalysis of the EC $CO_2$ data and derivation from existing gas transfer velocity parameteristions. The detailed procedures for determining and parameterising $\Delta_s$ using both methods are described in the "Methods" section, and here, we provide only a brief overview. Both approaches require prior knowledge of $\delta$ and $K_{int}$. This study adopts the recent estimate of $\delta$ for $CO_2$ from a bubble dynamic model ($\delta = 0.0132$[11]), and employs the $K_{int}$ parameterisation based on the EC DMS (dimethylsulfide) observations[19]. In the first method, we re-analyse the EC datasets to

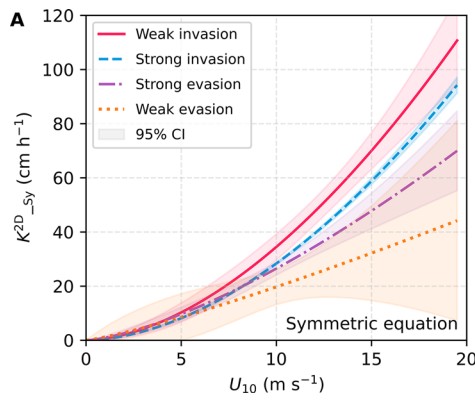

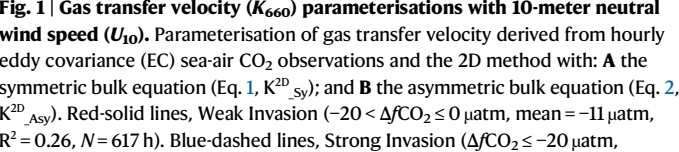

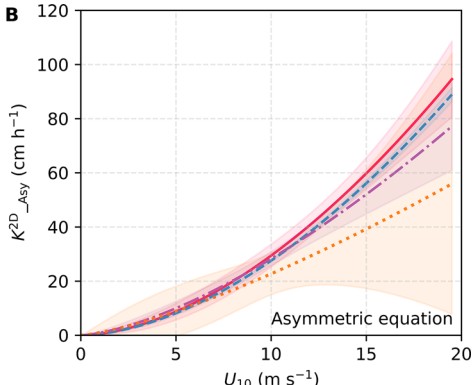

**Fig. 1 | Gas transfer velocity ($K_{660}$) parameterisations with 10-meter neutral wind speed ($U_{10}$).** Parameterisation of gas transfer velocity derived from hourly eddy covariance (EC) sea-air $CO_2$ observations and the 2D method with: **A** the symmetric bulk equation (Eq. 1, $K^{2D}_{Sy}$); and **B** the asymmetric bulk equation (Eq. 2, $K^{2D}_{Asy}$). Red-solid lines, Weak Invasion ($-20 < \Delta fCO_2 \leq 0$ µatm, mean = −11 µatm, $R^2$ = 0.26, N = 617 h). Blue-dashed lines, Strong Invasion ($\Delta fCO_2 \leq -20$ µatm,

mean = −68 µatm, $R^2$ = 0.63, N = 2889 h). Purple-dot-dashed lines, Strong Evasion ($\Delta fCO_2 \geq 20$ µatm, mean = 29 µatm, $R^2$ = 0.41, N = 236 h). Orange-dot lines, Weak Evasion ($0 < \Delta fCO_2 < 20$ µatm, mean = 9 µatm, $R^2$ = 0.014, N = 340 h). Here, the $R^2$ refer to the fits in (**A**); those for panel B are similar (see Table S1). The 95% confidence intervals (CI) are added to each parameterisation curve, using corresponding colours.

estimate $\Delta_s$, which is then fitted against wind speed (Fig. S5). This yields the following parameterisation:

$$\Delta_s = 0.0132\left(1 - 1.37U_{10}^{-0.37}\right), U_{10} \geq 5\,\text{m s}^{-1} \tag{5}$$

The alternative way to determinate $\Delta_s$ is by linking $\Delta_s$ with the fractional contribution of bubble-mediated gas transfer velocity to the total $K$ (see Eq. (4)). If the widely-used [14]C-based parameterisation[20] is adopted to represent the total $K$, $\Delta_s$ can be derived as:

$$\Delta_s = 0.0132(1 - 2.95U_{10}^{-0.67}), U_{10} \geq 5\,\text{m s}^{-1} \tag{6}$$

For wind speeds below $5\,\text{m s}^{-1}$, $\Delta_s$ is set to zero for both parameterisations because bubble contributions are negligible under this condition. The $\Delta_s$ values from Eqs. (5 and 6) diverge at wind speeds below $10\,\text{m s}^{-1}$, but they converge at high wind speeds ($10–20\,\text{m s}^{-1}$), with differences of less than 10%. Both parameterisations yield comparable results for the subsequent analysis within this section; therefore, only results based on Eq. (5) are presented in the figures below.

When the asymmetric equation (Eq. (2)) is used for the 2D fit, the invasion transfer velocity decreases, especially for the weak invasion group, while the evasion transfer velocity increases. $K^{2D}_{Asy}$ show much less divergence and are mostly collapsed onto a single curve (Fig. 1B). There is no statistically significant difference between $K^{2D}_{Asy}$ across the different flux regimes (Fig. S4B). The weak evasion group in Fig. 1B remains an outlier (lower $K^{2D}_{Asy}$ than the other three groups), which may well be attributable to the large relative uncertainty in these observations and fewer data points (8% of the total data points). $K^{2D}_{Asy}$ is less dependent on flux direction and magnitude, suggesting that the asymmetric model more consistently reflects the underlying physical processes across varying flux conditions. It is important to emphasize that the improvement offered by the asymmetric equation is not primarily demonstrated through a better statistical fit (e.g., $R^2$) to noisy field data, but rather through the reduction in the systematic divergence between the four $K^{2D}_{Sy}$ groups (as evidenced by the difference between Fig. 1A, B). The robustness of the 2D method is evaluated in detail in the "Methods" section.

These results support the use of the asymmetric equation (Eq. (2)) rather than the symmetric formulation (Eq. (1)) for interpreting EC observations and calculating bulk fluxes. Previous research has tended to use the symmetric equation to derive $K$ and then parameterise with wind speed using the 1D fit approach (i.e., $K^{1D}_{Sy}$)[21]. Our analysis shows that this method has overestimated $K$ (especially at high wind speeds,

Fig. S6) because most of the existing observations were collected under invasive scenarios. The bulk flux estimated using the asymmetric equation and the 2D fit method agrees better with observed EC $CO_2$ fluxes compared to bulk fluxes estimated using the conventional symmetric equation and the 1D fit method (Fig. S7), indicating that the asymmetric equation is more appropriate for bulk $CO_2$ flux estimates.

$K^{2D}_{Asy}$ based on all EC data is consistent with the $K_{660}$-$U_{10}$ parameterisation constrained by the global [14]C inventory[20] (Fig. S8). We note that $K_{660}$ derived from the [14]C inventory is insensitive to the asymmetric bubble transfer because the ocean is in large disequilibrium with respect to radiocarbon in the atmosphere[22].

The over-pressure factor ($\delta$) of 0.0132[11] is needed to determine $\Delta_s$. The small fraction of remaining divergence shown in Fig. 1B suggests that $\delta$ may be slightly underestimated. If $\delta$ is increased to 0.018, $K_{660\_CO_2}$ derived from the asymmetric bulk equation fully collapses the parameterisations for the weak invasion, strong invasion, and strong evasion groups (Fig. S9). However, uncertainty in the EC data could lead to overfitting, especially when using a small dataset (e.g., two evasion groups). The published value of $\delta$ = 0.0132 is thus used for the rest of this study, and Eqs. (5 and 6) are applied accordingly, as this value is based on independent evidence. If $\delta$ is better constrained in the future, Eq. (5 and 6) can be readily updated by replacing the coefficient 0.0132 with the revised value.

## Impact of asymmetry on large-scale $CO_2$ flux estimates

Accurate global sea-air $CO_2$ flux estimates are crucial for the Global Carbon Budget (GCB) assessment[2]. The GCB calculates sea-air $CO_2$ flux using the symmetric bulk equation, but previous results provide evidence of bubble-induced asymmetry in gas exchange[11–13,16], and our results further support that this asymmetry is important for sea-air $CO_2$ transfer. Here, we assess the impact of the asymmetric bubble transfer on global sea-air $CO_2$ flux estimates. The $CO_2$ flux from 1991 to 2020 is recalculated using the asymmetric bulk equation (Eq. (1)) and compared with the results using the symmetric bulk equation (Eq. (2)) (see "Methods"); their difference yields $\Delta Flux$ (i.e., asymmetry-induced flux). To ensure comparability, all flux estimates use the [14]C-based $K_{660}$-$U_{10}$ parameterisation[20], with coefficients scaled to the ERA5 wind speed[23]. Both $\Delta_s$ parameterisations are used for this global ocean assessment. The global mean value of $\Delta_s$ is estimated to be 0.004 (i.e., 0.4%) using Eq. (5) and 0.003 (0.3%) using Eq. (6).

The global ocean $CO_2$ uptake computed using the asymmetric equation is $0.33–0.41\,\text{Pg C yr}^{-1}$ greater than using the symmetric equation on average from 1991 to 2020, corresponding to ~15%

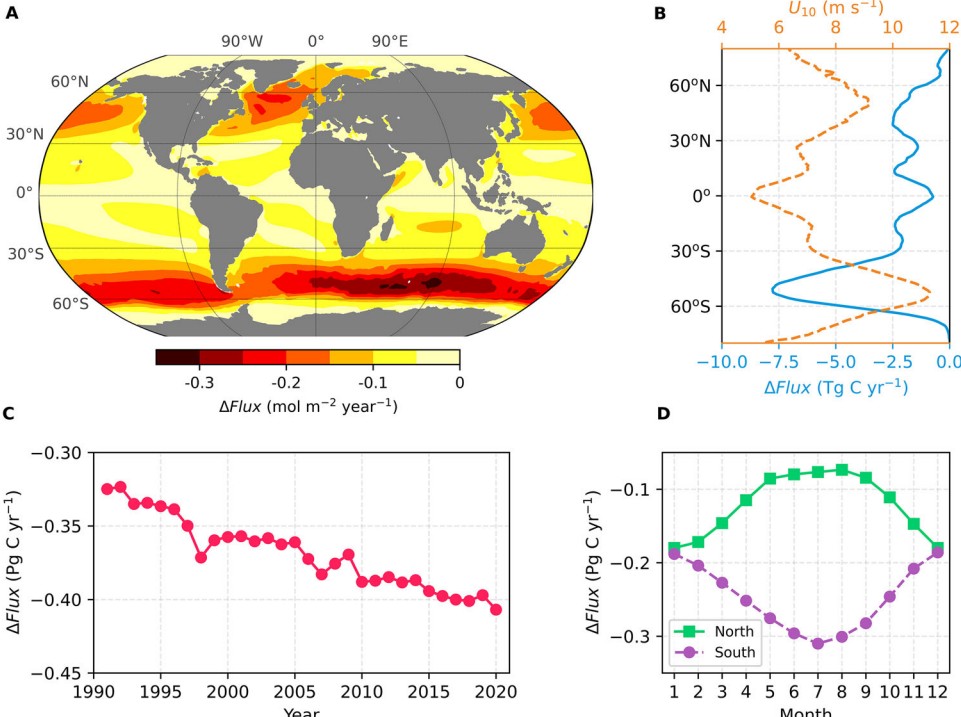

**Fig. 2 | Impact of asymmetric transfer on the sea-air $CO_2$ flux estimate (ΔFlux).** **A** Map of ΔFlux; **B** 1°-latitude mean of ΔFlux and ERA5 wind speed; **C** Temporal trend in annual mean ΔFlux; **D** Seasonal variations of ΔFlux in northern (green) and southern (purple) hemispheres (1–12 corresponding to January-December). The results shown here represent the ensemble mean ΔFlux estimated from two different $\Delta_s$ parameterisations (Eqs. 5 and 6). The ΔFlux shown in (**A**, **B**, and **D**) is averaged from 1991 to 2020. A negative ΔFlux means enhanced ocean $CO_2$ uptake.

increase in the oceanic $CO_2$ sink estimates. Equation (5) produces higher $\Delta_s$ under typical oceanic wind conditions (5–10 m s$^{-1}$; Fig. S5) and thus yields a larger ΔFlux magnitude (0.41 Pg C yr$^{-1}$) than Eq. (6) (0.33 Pg C yr$^{-1}$). This difference highlights the uncertainty associated with quantifying $\Delta_s$. The impact of the asymmetry on sea-air $CO_2$ flux is ubiquitous, but is most evident in the Southern Ocean (South of 35°S) and relatively minor in the tropics (Fig. 2A). The Southern Ocean accounts for about half of the asymmetry-induced flux increase in the global ocean. The spatial variability of ΔFlux is primarily driven by wind speed (Fig. 2B), as stronger winds enhance wave breaking and bubble formation, thereby amplifying asymmetric bubble-mediated transfer. Notably, the impact of ΔFlux is always negative (i.e., enhanced ocean $CO_2$ uptake) because the bubble over-pressure always favours gas invasion. Over the past three decades, ΔFlux has shown a strengthening trend in a rate of ~3 Tg C yr$^{-1}$ per decade (Fig. 2C). This trend is primarily driven by the rising atmospheric $CO_2$ concentration. In addition, hemispheric ΔFlux varies seasonally, with greater asymmetrical fluxes in winter and smaller fluxes in summer (Fig. 2D). The seasonal variability is primarily driven by seasonal wind variation and sea surface temperature changes.

Note that Fig. 2 does not consider the upper ocean temperature effects[24] in the calculation of global sea-air $CO_2$ flux. Recent studies provide relatively direct evidence that the cool skin effect does affect the bulk sea-air $CO_2$ flux estimates[25,26]. The cool skin effect has been estimated to increase oceanic $CO_2$ uptake by ~0.4 Pg C yr$^{-1}$ [27,28]. All previous cool skin studies apply the correction to the total gas transfer velocity, but the cool skin effect theoretically only influences interfacial transfer, whereas bubbles bypass the cool skin. We have re-evaluated the cool skin correction by only considering interfacial transfer (see "Methods"). Our results suggest a smaller cool skin correction ($CO_2$ uptake increase by ~0.25 Pg C yr$^{-1}$ on average from 1991 to 2020, ~2/3 of previous estimates). The Surface Ocean $CO_2$ Atlas

(SOCAT)-based flux in the 2023 GCB using the symmetric flux formula suggests that ~65% of the global surface ocean has net $CO_2$ invasion[2]. Applying asymmetric transfer along with updated temperature effects increases the area of net invasion to ~75% (Fig. S10) regardless of whether $\Delta_s$ is estimated using Eqs. (5 or 6). The change in sign from evasion to invasion primarily occurs in the high-latitude Southern Ocean and in oligotrophic waters. The updated climatological flux estimate shows that the global ocean is generally a $CO_2$ sink, with $CO_2$ outgassing only occurring in regions with upwelling (e.g., near the equator and the coast, Fig. S10).

The GCB reports global ocean $CO_2$ uptake using both SOCAT-based contemporary flux estimates and Global Ocean Biogeochemistry Models (GOBMs)-based anthropogenic fluxes[2]. The anthropogenic ocean $CO_2$ uptake is defined as the contemporary net sea-air $CO_2$ flux adjusted for the riverine $CO_2$ flux to the ocean (0.65 Pg C yr$^{-1}$ with large uncertainties[29]). Note that the asymmetric effect and the updated cool skin-induced flux corrections correspond to SOCAT-based flux, which cannot be directly applied to GOBMs[30]. Both the cool skin effect and the asymmetric transfer effect increase the net invasion flux in the upper mixing layer, but only a fraction of the additional $CO_2$ can be transported to the deeper ocean in the model because of the slow vertical ocean circulation. This results in an accumulation of carbon in the mixing layer (i.e., an increase in $C_w$) and thus dampens the flux enhancement. It has been estimated that ~2/3 of the impact of the cool skin effect on global $CO_2$ flux will be dampened within a GOBM[30], and we assume the same damping magnitude for the impact of the asymmetric transfer. After accounting for asymmetric transfer and updating the cool skin effect as well as incorporating another temperature correction (warm bias[28]), GOBMs-based ocean $CO_2$ uptake is ~2.4 Pg C yr$^{-1}$ (on average from 1991 to 2020), which is 1 Pg C yr$^{-1}$ (30%) lower than the SOCAT data-based estimates (Table 1).

## Discussion

This study uses EC sea-air $CO_2$ flux observations with both invasion and evasion scenarios to present direct evidence that asymmetric bubble-mediated transfer is significant for $CO_2$ exchange, especially at high wind speeds (Fig. 1). The evidence is broadly in line with the concepts proposed in a previous study[11]. The asymmetric bulk equation (Eq. (2)), with $\Delta_s$ from Eqs. (5 or 6), is recommended for sea-air $CO_2$ flux estimates and for EC sea-air $CO_2$ flux-based $K_{660}$ analyses. Published EC-based $K_{660}$ data (e.g., a synthesis study[21]) contain biases due to the use of the symmetric bulk equation to derive $K_{660}$. The bias is larger for cruises with high wind speeds and weak invasion/evasion fluxes[19,31], and smaller for cruises with strong invasion/evasion flux signals and low-medium wind speeds[32]. The observed asymmetry is further evidence that bubble-mediated transfer is important for sea-air $CO_2$ flux, consistent with the large differences between the gas transfer velocities of $CO_2$ and DMS[19,33–35] and the sea state dependence of $CO_2$ transfer velocities[36].

Using the constrained asymmetric factor, the asymmetric effect results in an additional oceanic $CO_2$ uptake of 0.3–0.4 Pg C yr$^{-1}$ (1991 to 2020 average) compared to the uptake calculated with the symmetric bulk equation. The asymmetric flux has wind-driven regional and seasonal variations, and is relatively large in the Southern Ocean and during winter (Fig. 2). The influence of asymmetric bubble transfer on sea-air $CO_2$ flux has increased over the past decades due to ever-rising atmospheric $CO_2$ concentration (Fig. 2C). The revisions to global climatological $CO_2$ flux increase the ocean areas with net $CO_2$ invasion from ~65% to ~75%, leaving only the upwelling regions with net $CO_2$ evasion. The revisions also widen the gap between the SOCAT-based flux estimates and the GOBMs-based flux estimates (from 0.4 Pg C yr$^{-1}$ to ~1.0 Pg C yr$^{-1}$). Reconciling the difference between model-based and SOCAT data-based sea-air $CO_2$ flux estimates is a major challenge to the community. Resolving possible model biases due to inadequate simulation of ocean circulation and oceanic buffer capacity has been proposed[37]. With respect to the observations, the sparsity of SOCAT data has been identified as a major source of uncertainty in SOCAT-based sea-air $CO_2$ flux estimates[38]. Moreover, reducing the uncertainties associated with the riverine flux is also critical for understanding the discrepancy between model and data-based flux estimates[39].

This study provides observational evidence of asymmetric $CO_2$ transfer using a large dataset ($N = 4082$ h). The EC sea-air $CO_2$ flux dataset is dominated by measurements in net invasion conditions (86%, $N = 3506$ h), whereas there are fewer net evasion observations ($N = 576$ h), which limits our confidence in the global asymmetry-adjusted ocean $CO_2$ uptake estimate. $\Delta_s$ estimates from two different approaches are similar under high wind speeds ($U_{10} > 10$ m s$^{-1}$), but differ substantially at lower wind speeds. This difference results in large variations in the estimated impact of bubble-induced asymmetry on global ocean $CO_2$ uptake, highlighting the need to reduce uncertainties in the $\Delta_s$ estimates. Nevertheless, the value of $\Delta_s$ (0.3–0.4% on average) estimated in this study is consistent with existing evidence. Field noble gas observations indicate Xenon (Xe) supersaturation of ~1% under typical ocean conditions[12]. The solubility of Xe ($\alpha$ ~ 0.1 at 20 °C) is lower than that of $CO_2$ ($\alpha$ ~ 0.7 at 20 °C), meaning that the $\Delta_s$ of $CO_2$ is expected to be less than 1%. Another independent estimate uses a bubble dynamic model designed for low solubility gases, and extrapolates $a$ ~ 0.7% supersaturation factor for $CO_2$[13]. Still, more direct sea-air $CO_2$ flux measurements are needed to reduce the uncertainty associated with the bubble-induced supersaturation factor, and strengthen and improve the asymmetric parameterisations proposed here. Future observations should target $CO_2$ evasion as a priority at high wind speeds and over a wide range of sea states. A mixture of methodologies that encompass evasion, invasion, and a range of gases with different solubilities would provide even stronger evidence of asymmetric bubble-mediated transfer (e.g., wintertime in the Bering Sea, or the summer monsoon season in the Arabian Sea). In the long term, expanding EC sea-air $CO_2$ flux observations using autonomous platforms such as a buoy[40], Saildrone, and/or Wave Gliders will provide an essential reference for bulk flux estimates.

## Methods

### Two-dimensional analysis of the $CO_2$ flux

A recent study[21] presents a synthesis of high-quality EC sea-air $CO_2$ flux and $\Delta fCO_2$ measurements made over the last ~15 years (2698 h). These data were collected from 11 research cruises conducted in the North Atlantic Ocean[17,19,34], the Southern Ocean[35,41,42], the Arctic Ocean[32], and the Tropical Indian Ocean[31]. There were both net invasion and net evasion observations in this synthesis dataset ($\Delta fCO_2$ ranges from −273 μatm to 76 μatm). A further six EC sea-air $CO_2$ flux and $\Delta fCO_2$ datasets are included in this analysis, two during the Atlantic Meridional Transect cruises (732 h)[18] and four in the Southern Ocean (652 h)[25]. All of the datasets (17 research cruises, see Fig. S1) are combined to investigate the bubble-induced asymmetry. The EC system setup for different cruises, data quality control, and data processing are presented in a synthesis study[21] and the literature referenced therein. The Atlantic Ocean and the Southern Ocean datasets are described in related literatures[18,25].

The EC-based $K$ is traditionally computed as "EC flux/ $\alpha\Delta fCO_2$" and then fitted with wind speed after Schmidt number normalisation (i.e., one dimensional fitting method, $K^{1D}$). However, $K$ derived in this way becomes unreliable when $\Delta fCO_2$ is close to 0, and thus data with small absolute $\Delta fCO_2$ (typically |$\Delta fCO_2$|<20 μatm) are often excluded from analysis. The excluded near-saturation data are useful because the influence of asymmetric transfer is expected to be relatively large (see Eq. S4). A recent study demonstrated that EC flux observations are still reliable even when the sea-air $CO_2$ flux is ~0 and the small EC fluxes often contain small absolute uncertainties[18]. To make use of the low flux signal data, an alternative two-dimensional (2D) fit method is employed for analysis. Rather than fitting the derived $K$ as a function of wind speed, flux data are fit as a function of both concentration difference and wind speed with the following functional structure:

$$Flux = \Delta C_{660}(aU_{10}^b) \tag{7}$$

where $\Delta C_{660}$ is equal to $(C_w - C_a)(Sc/660)^{-0.5}$ if using the symmetric bulk equation, and $[C_w - (1 + \Delta_s)C_a](Sc/660)^{-0.5}$ if using the asymmetric bulk equation. The wind speed dependence of the gas transfer velocity with the 2D fit ($K^{2D}$) has an assumed structure, with free parameters "$a$" and "$b$". The fit is to the flux, meaning that the error minimisation is on the predicted flux (i.e., a "least squares" fit to flux; see Supplementary Information, Section 2).

The EC data is separated into four groups according to $\Delta fCO_2$ (see the caption of Fig. 1). The 2D fit is applied to each data group, and also to a combined group of strong evasion and strong invasion data, and to the entire dataset. The direct 1D fit between $K_{660}$ and $U_{10}$ is only applied to the strong evasion and invasion groups, as well as the combined group containing strong evasion and invasion data. Coefficients "$a$" and "$b$" and the $R^2$ for each fit are reported in Table S1.

### Estimation of the asymmetry factor $\Delta_s$

The asymmetry factor ($\Delta_s$) in Eq. (2) is a key parameter in this study. We estimate $\Delta_s$ using two approaches. Both methods rely on the independent estimates of the over-pressure factor ($\delta$), the interfacial transfer velocity ($K_{int}$) and the total gas transfer velocity ($K$) (see Eq. 4). For $CO_2$, $\delta$ is primarily driven by the hydrostatic pressure and is directly related to the effective penetration depth of the bubble plume, which has been shown to remain largely unchanged with wind speed[9,43]. Accordingly, we adopt a fixed $\delta$ value of 0.0132, simulated from a bubble dynamic model-based on near-surface bubble observations[11]. For $K_{int}$, we use transfer velocity parameterisations based on EC DMS observations[19] (Fig. S8), as the high solubility of DMS

**Table 1 | Corrections and revisions to the estimate of global ocean anthropogenic $CO_2$ uptake**

| Oceanic $CO_2$ uptake estimates | GCB 2023 | Corrections | | | Revised flux |
|---|---|---|---|---|---|
| | | Asymmetric effect | Cool skin effect | Warm bias | |
| Based on SOCAT data | 2.60 | 0.37 | 0.25 | 0.17 | 3.39 |
| Based on models | 2.23 | 0.12 | 0.08 | – | 2.43 |

Two independent oceanic $CO_2$ uptake estimates are made in the Global Carbon Budget (GCB) 2023[2]. The values in the "asymmetric effect" column represent the ensemble mean of the flux corrections using two $\Delta_s$ parameterisations (Eqs. 5 and 6). The correction to the model-based flux estimate is assumed to be ~1/3 of the correction to the data-based flux estimate[30]. The warm bias correction is taken from ref. 28. All numbers in the table are in Pg C yr$^{-1}$ and represent the average from 1991 to 2020.

minimises the contribution of bubbles to its exchange. Based on the way to represent $K$, two approaches are proposed to estimate $\Delta_s$.

In the first approach, hourly EC sea-air $CO_2$ flux measurements are treated as the total $CO_2$ exchange in the estimate of estimate $\Delta_s$. However, these EC $CO_2$ fluxes inherently include the effect of chemical enhancement (CE), which is absent in EC DMS observations that are being used for $K_{int}$. While CE is negligible at high wind speeds, it becomes relatively important under low wind conditions. Given that $\Delta_s$ is sensitive to the treatment of $K$ at low wind speeds, CE contribution should be removed from the observed EC $CO_2$ transfer velocities. To account for this, we applied a CE correction ratio derived from analytical and numerical models[6,44]. We then combine Eqs. (2 and 4) to derive $\Delta_s$ for each corresponding flux using an iterative method (see Supplementary Information, Section 3). The resulting $\Delta_s$ increases with the wind speed and asymptotically approaches the value of $\delta$ when the bubble-mediated exchange dominates the total gas exchange (Fig. S5). Given that $\Delta_s = \delta K_{bub}/K = \delta(K - K_{int})/K$, and $K$ and $K_{int}$ can be expressed as $a_1 U_{10}^{b1}$ and $a_2 U_{10}^{b2}$, respectively, $\Delta_s$ can thus be parameterised as $\Delta_s = \delta(1 - a_3/U_{10}^{b3})$. Fitting the bin averages of $\Delta_s$ for $U_{10}$ between 5 and 20 m s$^{-1}$ yields Eq. (5).

Alternatively, the widely used $^{14}C$ inventory-based parameterisation[20] can represent the total $K$ (Fig. S5A). Thus, $\Delta_s$ can be directly calculated using the existing parameterisations of $K$ and $K_{int}$, which yields Eq. (6). If the recent $K$ parameterisation based on the synthesis of the EC sea-air $CO_2$ data[21] is used to represent the total gas transfer velocity, the derived $\Delta_s$ will be similar to Eq. (5) (Fig. S5B), and thus is not shown in the main text. It should be noted that the $^{14}C$ inventory-based parameterisation of $K$ does not include chemical enhancement[20] and therefore require no correction, whereas the $K$ parameterisation derived from EC $CO_2$ data[21] does include this chemical effect and thus necessitates a correction.

### Robustness test of the 2D fitting method

To assess the robustness of the 2D fitting approach, we performed several sensitivity tests.

First, we applied a bootstrapping test by systematically excluding one or more cruises from the entire dataset (Fig. S11). The divergence between invasion and evasion groups observed in the symmetric equation consistently merges (Fig. S11A), and the asymmetric equation continues to reduce this divergence across all subsets (Fig. S11B). This consistency demonstrates that the observed pattern is not driven by a few specific cruises or one specific research group, and supports the stability of the asymmetric equation and the 2D fitting method. We note that the weak evasion is always an outlier due to less data and high uncertainty as shown in Fig. 1 and discussed in the main text.

Second, the results shown in Fig. 1 do not consider the cool skin effect, which can also lead to bias in the derived $K_{660}$. However, the cool skin effect is relatively more substantial at low wind speeds and relatively weak and consistent at intermediate and high wind speeds[45]. Inclusion of the cool skin effect in the derivation of $K^{2D}$ does not collapse any divergences shown in Fig. 1A (see Fig. S12). Other processes, such as sea spray and rainfall (both of which often occur during stormy conditions), may also induce asymmetric gas transfer. Sea spray tends to enhance evasion relative to invasion[46], which is opposite to the pattern observed in Fig. 1. Sea spray-related asymmetries remain highly

uncertain[47] and are beyond the scope of this study. Rain events may also promote asymmetric $CO_2$ uptake[48,49], but rain intensity is generally independent of wind speed and thus unlikely to explain the wind speed-dependent divergence observed in Fig. 1.

For the 2D fit, we limit the form of $K_{660}$ to $aU_{10}^b$ (Eq. 7) with zero intercept, and it was plausible that this form forced most of the divergence into the high wind speeds. To check this, we relaxed the constraint in Eq. 4 and adopted a more flexible formulation ($aU_{10}^b + c$), allowing for a non-zero intercept. The results confirm that the collapse between invasion and evasion at high wind speeds persists (Fig. S13), further supporting the robustness of the 2D fitting approach.

### Global ocean $CO_2$ flux estimates

The global ocean $CO_2$ flux is estimated using the asymmetric bulk equation and the symmetric bulk equation. The difference between these two fluxes is considered the additional flux due to asymmetric bubble-mediated transfer. The global bomb-$^{14}C$ inventory-based $K_{660\_CO2}$ parameterisation[20,22] is used to make the flux estimate. $\Delta_s$ in Eq. (2) is estimated from ERA5 wind speed[23] when $U_{10}$ is higher than 5 m s$^{-1}$ (Fig. S5) and set as zero when $U_{10} < 5$ m s$^{-1}$ since bubble-mediated transfer should be minimal at low wind speed[34]. For the revised cool skin correction, the $K_{int}$ parameterisation based on the EC observations of DMS transfer[19] (Fig. S8) is used to calculate the interfacial $CO_2$ flux. For the interfacial flux with cool skin correction, $\Delta C$ is calculated as $\alpha_{subskin} fCO_{2w} - \alpha_{skin} fCO_{2a}$, while the flux without cool skin correction uses $\Delta C = \alpha_{subskin} fCO_{2w} - \alpha_{subskin} fCO_{2a}$. Here, $\alpha_{subskin}$ and $\alpha_{skin}$ are the $CO_2$ solubility calculated using subskin and skin seawater temperature, respectively[28]. ERA5 wind speed data from 1991 to 2020 are used to estimate the transfer velocity for the global ocean at a 1° × 1°, monthly resolution. The ensemble mean of seven SOCAT-based $fCO_{2w}$ products (1° by 1°, monthly)[2] is used as the $fCO_{2w}$ product. Global atmospheric $CO_2$ fugacity ($fCO_{2a}$) data is calculated from NOAA ESRL marine boundary layer $CO_2$ mole fraction[50]. The CCI SST v2.1 data product[51] is used to estimate Schmidt number[20] and $\alpha_{subskin}$[52] for the global ocean.

### Data availability

All data needed to evaluate the results in the paper are present in the paper and/or the Supplementary Information. SOCAT-based data products and Global Ocean Biogeochemistry Models: https://zenodo.org/records/10222484; ERA5 wind speed: https://cds.climate.copernicus.eu/cdsapp#!/dataset/reanalysis-era5-single-levels?tab=form; The data to directly produce Figs. 1 and 2 are provided in the Supplementary Information/Source Data file. The reanalysed EC data generated in this study have been deposited in the Figshare database under accession code: https://doi.org/10.6084/m9.figshare.29903636.

### Code availability

The code to produce the figures are provided in the Supplementary Information/Source Data file.

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

## Acknowledgements
Y. Dong has been supported by the Alexander von Humboldt Foundation. M. Yang and T. Bell have been supported by NERC (ORCHESTRA, NE/N018095/1, and PICCOLO NE/P021409/1 projects) and the European Space Agency (AMT4oceanSatFluxCCN, 4000125730/18/NL/FF/gp). D. Woolf thanks the European Space Agency for funding and support under the "Ocean Carbon for Climate" project (OC4C; 3-18399/24/I-NB). The Python programming language was used to analyse the data and generate all the figures in the manuscript. Additionally, the Python-Basemap package was utilized to create Figs. 2A, S1A, and S10A, B.

## Author contributions
Y.D. and T.G.B. came up with the initial idea. M.Y., D.K.W., T.G.B. and Y.D. conceived the study. M.Y., C.A.M. and T.G.B. provided most of the EC data. D.K.W. contributed to the reanalysis method of the EC data, and M.Y. proposed the 2D analysis method. Y.D. and M.Y. performed the data analysis with help from all other authors. Y.D. wrote the initial draft, and all coauthors contributed to the writing.

## Funding

## Competing interests
The authors declare no competing interests.
