## [Transparent Peer Review file · Nature Communications]

Asymmetric bubble-mediated gas transfer enhances global ocean CO₂ uptake

Corresponding Author: Dr Yuanxu Dong

Version 0:

Reviewer comments:

Reviewer #1

(Remarks to the Author)

The authors use eddy covariance data to discuss the importance of an asymmetric contribution in air-sea CO₂ fluxes. Eddy covariance data which provide the flux and the partial pressure difference are used to infer an asymmetric parameterization. The authors argue that accounting for such asymmetric effect improve the match to the data. They then use the new parameterization to estimate global CO₂ fluxes using other global observation based partial pressure difference products, and find a significant different with numbers currently in the literature.

In my opinion, the paper presents significant issues in terms of presentation, methodology and the subsequent interpretation.

The authors state that they use a “2D fit” to demonstrate the differences in evasion and invasion configurations, to demonstrate that the symmetric only formulation fails.

A major problem to me is that the authors do not show their data in the main manuscript. They only show a resulting fit to the data in figure 1A which they use as proof of their argument. This is not acceptable as the reader should see the empirical evidence from their argument and assess whether the fit is improved when asymmetric contribution is considered.

The data are shown in supplementary and it is not very convincing. When I look at figure S4, it is very difficult for me to get convinced that your symmetric fit and asymmetric fit are different in a statistical way. There is so much noise around the fitted line.

Similarly, figure 1B is not convincing if you do not show the data.

Line 158: the authors state: “but we have demonstrated that asymmetric bubble 159 exchange is important for sea-air CO₂ transfer”. I am sorry but they have not demonstrated anything. They are stating something but the empirical evidence is not shown in a convincing way.

Next, I would focus on data at high wind speed, there are not much bubbles at 5 m/s so it is unclear why they are considered. I would use at least 10 or 15 m/s as cut-off. You will have less data but maybe better quality. Sensitivity to the amount of data used need to be discussed.

I would suggest plots similar to the one presented in Stanley et al 2022, where a supersaturation is evaluated, which the authors could estimate through a ratio of water and air side concentration or partial pressure. They could then demonstrate that for the same partial pressure difference, the flux is much larger in the case where bubbles do play a role (invasion) than in evasion configuration. The authors need to find a way to demonstrate the claim in their data, which is not done in the present version of the manuscript. I would insist that the work from Stanley (and others) clearly demonstrate the existing of the asymmetric term in the flux.

Also, there are some existing asymmetric parameterization in the literature, Stanley et al 2009 is one; Liang et al 2013 is another; and while they were presented for other gases, the authors might be able to compare to their proposed version

since they include some dependency in diffusivity and solubility.

Moving to the global flux estimates, I think the discussion is questionable. Of course if you take the same Δp_{CO_2} data and change the k_w formulation, you will get large differences. But what if you had implemented your asymmetric formulation in a global ocean model? Would the model equilibrate to similar flux after some transient? This has been the argument of the modeling community to keep using the Wanninkhof quadratic fit.

Reviewer #2

(Remarks to the Author)

April 3, 2025

Review of: Asymmetric bubble-mediated gas transfer enhances global ocean CO₂ uptake, by Dong et al.,

This paper argues that the very impressive results of air-sea CO₂ exchange fluxes from Eddy Correlation (EC) measurements at sea indicate that there is evidence for an asymmetric exchange flux of CO₂ at the air-water interface due to the enhanced invasion rate from bubble fluxes created by wave breaking. Background arguments in the “Materials and Methods” and in the “Supplemental Information” present a strong case that the CO₂ EC flux method has resulted in gas exchange coefficient, k_{660} , values as a function of wind speed that are broadly consistent with values determined by other methods based on bomb ¹⁴C and gas tracer release experiments. Furthermore, the great difference between k_{660} determined with CO₂ and DMS eddy correlation experiments suggest that bubble exchange plays a significant role in CO₂ air-sea exchange. This is a meaningful review of the existing data that have been published (e.g., Yang et al., 2022; Leighton et al., 2018).

The point of this revisit to the CO₂ EC results is to make the case that evidence for a bubble effect on the CO₂ gas exchange rate exists in the compiled data from field experiments. The authors make the argument that CO₂ EC data cannot be successfully interpreted as entirely due to symmetric gas exchange in which global air-sea transport by CO₂ invasion and evasion are caused by the same mechanism. Rather, they suggest a model in which there is a small excess CO₂ invasion flux that is due to the transport by bubbles. It is well known that the bubble fluxes during air-sea exchange of the highly insoluble atmospheric gases of N₂, O₂, and Ar cause measurable supersaturation in surface waters. CO₂, however, is much more soluble than these gases and the role of bubbles is more difficult to evaluate.

The authors argue that one has to use a model that accounts for asymmetric air-sea exchange to interpret the existing data. The asymmetric factor is the supersaturation, Δs , caused by bubble fluxes. If Δs is zero the bubble flux is zero, and as you increase Δs the evidence for the importance of bubbles increases to the point where it makes a significant difference in the calculation of the amount of anthropogenic CO₂ that has entered the ocean, which the authors do convincingly.

I am impressed by this paper: the authors have written it well, and I believe they have done an impressive job of interpreting the marine CO₂ EC data, which is a challenge because there is a lot of scatter. However, I do not think they have made a strong case, that, as stated in the abstract and conclusions, “This study provides direct field evidence of asymmetric bubble-mediated transfer...” They show that evidence for the importance of bubble-mediated CO₂ exchange in their analysis depends strongly on the value of Δs , and it is pretty clear that they are uncertain about what this value is. I think this is a major flaw in the paper and makes one skeptical of the conclusions about the anthropogenic fluxes. I would recommend that, at the very least, the authors do an error analysis (something like a Monte Carlo uncertainty calculation) to determine how well one needs to know Δs to be able to constrain the importance of bubbles in the interpretation of air-sea CO₂ exchange.

Reviewer #3

(Remarks to the Author)

Review #1 of Dong Y., M. Yang, T. G. Bell, C. A. Marandino, D. K. Woolf: Asymmetric bubble-mediated gas transfer enhances global ocean CO₂ uptake submitted to Nature Communications (2025)

The article is well written and is addressing very important and long-lasting issue on the global impact of bubble-mediated CO₂ air-sea exchanges. To do so, the authors propose an original approach based on the re-analysis of a large amount of in situ Eddy Covariance (EC) flux measurements. The authors showed that assuming an asymmetric model for the CO₂ flux better explain the in-situ measurements. New values for the parameters are determined to provide an estimate of the global impact of bubble-mediated CO₂ transfer. The impact of this process is to increase the global ocean CO₂ sink by about 25%. The authors also re-assess the impact of the ocean cool skin on CO₂ exchanges that is only applicable to the interfacial part of the flux and not to the bubble-mediated one. Finally, the authors discuss the fact that taking into account this process increases the disagreement between model-based and observational product-based assessments of the global ocean CO₂ sink. Since, ocean is a major compartment of the terrestrial Carbon cycle, this study is therefore likely to be of interest to a wide audience. The approach seems appropriate to back the results although some details on their statistical robustness are lacking (see below). Multiple references to the “Methods” and “Supplementary material” sections make it hard to follow. Working on the structure of these annexes and pointing specifically to sub-sections in the main text could help the reader.

Analytical approach:

Measurements from many campaigns are used in this study. This means that a diversity of sensors, a diversity of post-treatments of these data and on the top of this a diversity of physical conditions are used in this analysis. In addition, all these differences are not equally represented in the total ensemble and may not be independent: For example, all the weak

evasion situations might all be taken from a single campaign and suffer from a common experimental bias. How is this taken into account?

- The authors separate the EC data by invasion/evasion conditions but the information of the amount of data for each condition and on their origin among the 17 campaigns are lacking. This could be achieved by presenting the $K - U_{10}$ scatterplots for the 4 conditions that are defined in this study or providing histograms of number of measurements as a function of U_{10} and $\Delta_f CO_2$.

- Quality of the 2D fit should be characterized. In addition, robustness of the difference between symmetric and asymmetric models for K should be tested. This could be done with some kind of bootstrapping method, for example, by rejecting one or few campaigns data at a time.

Clarity and context:

The article is clear and well written. Here are some suggestions to improve the presentation of the context:

- In addition to bubbles and cool skin that are discussed here, other processes may be sources of asymmetries in the air-sea CO_2 transfer: Sea sprays which occur in the same conditions as bubbles and rainfall that tends to increase the ocean CO_2 uptake. The authors should discuss these processes for their results based on the recent literature.

Some suggestion to improve the clarity of the article:

- There are too many notations for the transfer velocity (sometimes not defined as $K_{CO_2_true}$), they should be homogenized maybe with symmetric/asymmetric in subscript, $obs/1D/2D$ in superscript and drop the references to the Schmidt number value and to CO_2 as they do not bare any useful information.

- As presented in the "Materials and Methods" section, the link between the asymmetry and the over-pressure factors is not clear.

Minor suggestions:

- Line 53: can you explain why bubble mediated transfer dependance on solubility is due to bubble finite volume?

- Lines 100-104, maybe the inclusion of equation (S4) could help the discussion here.

- Can the statistical significance in the different gas transfer velocities obtained for different situations be tested? Can this information be added in Figure 1?

- Supplementary Material line 61, what is the reference for the Kint parametric expression that is used here?

- The reference to Supp. Mat. should precise the number of the section where details are provided.

- EC fluxes re-analysis and fitting of K seems to be made on hourly means but this is never clearly stated.

- Lines-139-141: Why is a 1D fit used here instead of the 2D fit?

- Lines 161 and 238-240: The authors should clearly provide the final expression for K_{asy} (and Δ_s) that is recommended as equation (3).

- Lines 197-200: It has been shown that depressions in the Southern Ocean can induce outgassing due to decreased atmospheric pressure and entrainment of DIC-rich waters at the base of the mixed layer. This is not taken into account here and could reduce the impact of asymmetric transfer. Please comment.

Version 1:

Reviewer comments:

Reviewer #3

(Remarks to the Author)

Review #2 Dong Y., M. Yang, T. G. Bell, C. A. Marandino, D. K. Woolf: Asymmetric bubble-mediated gas transfer enhances global ocean CO_2 uptake, submitted to Nature Communications (2025)

Using a large collection of in situ eddy covariance measurements, the authors show that assuming an asymmetric model to take into account the impact of bubbles-mediated CO_2 exchanges better explain the in-situ measurements. This is evident as the transfer velocity coefficient obtained from an asymmetric formulation is less dependent on the CO_2 concentration difference across the air-sea interface. The new parameterization is used to calculate the global ocean CO_2 sink that is about 25% larger than previous estimates. The authors also re-assess the impact of the ocean cool skin on CO_2 exchanges

that is only applicable to the interfacial part of the flux and not to the bubble-mediated one.

The first version of the manuscript was suffering from (1) a lack of details on the representativity of the data and a lack of statistical robustness assessment of the fits and (2) a lot of different notations, a complex structure and a lack of reference to specific subsections of supplementary material. Overall, the authors addressed these two main drawbacks in a convincing way. They in particular provide a lot more details on the data and on the statistical significance of the change when considering an asymmetric formulation. Details are given below.

(1) Analytical approach

- In order to show that the dataset used encompasses a large variety of situations, and that each condition is documented by independent cruises, the authors now provide an overview of the data obtained in different conditions of CO₂ gradient and wind speed (Figure S1 and S2). It has yet to be noted that, on Figure S2, it is hard to clearly see the relative sampling of the different categories of conditions (weak/strong invasion/evasion and wind speed). A way could be to provide a 2D histogram (number of hours as a function of $\Delta f\text{CO}_2$ and U₁₀) or to break the histogram bars of Fig S2B for the different weak/strong invasion/evasion cases.

- The new version of Figure 1 provides new information on the uncertainty behind the fit. A statistical test is now performed (Figure S4) that shows that the use of an asymmetric formulation makes the transfer coefficients for different conditions (i.e. weak/strong evasion/invasion) statistically indistinguishable, as it should be.

- The robustness of the findings is further tested by performing a bootstrap test consisting in excluding cruises from the complete dataset. The authors show that the same conclusions can be drawn for each case (Figure S11). This result could maybe be made more quantitative and synthetic by defining a metric of the spread of the transfer coefficients functions for each panel of Fig S11 and then compare statistically the two ensembles shown on S11A and S11B.

(2) Clarity of the manuscript

The new manuscript is clear and the supplementary material subsections are referenced clearly in the text. Yet, I still have some comments that could help make the manuscript easier to follow:

- Some supplementary figures are not or hardly commented: Fig S1B and S1C, S3.

- Table S1 that has many panels that are hardly exploited, are they all necessary?

Minor comments

The authors finally considered the minor comments I made on the previous version and adapted the manuscript accordingly.

- I would like to suggest to the authors to provide the line numbers in their reply so that reviewers can efficiently track the changes.

- Line 118: "K1D_Asy" not "_Sy"

Reviewer #4

(Remarks to the Author)

Review of "Asymmetric bubble-1 mediated gas transfer enhances global ocean CO₂

Uptake" by Dong et al.

Summarization:

This study investigates bubble-mediated air-sea CO₂ exchange using eddy-covariance observations from 17 historical cruises. The authors develop an asymmetric parameterization scheme for CO₂ flux, demonstrating that bubble processes enhance oceanic uptake. Applying this scheme at the global scale, they estimate that the ocean absorbs an additional ~0.5 Pg C yr⁻¹ (over 25% increase) compared with conventional symmetric formulations, highlighting the substantial impact of bubble asymmetry on the global carbon budget.

I mainly have some concerns about implementing asymmetric equations in global ocean models, given the limitations of both the methodology and data.

Major Comment:

1. The effectiveness of this 2D asymmetric method is substantiated by the reduced divergence of K under weak evasion, strong invasion, and strong evasion conditions. However, this alone is not convincing evidence, as incorporating U₁₀ linearly into Δs then asymmetric equation will, by design, reduce the disparity in Figure 1(a) by extra polynomial terms.

Meanwhile, the new weighting method indeed cuts off the influence of low ΔC cases in another way and could artificially overestimate K or Δs with flux/wind speed increased. In my view, the authors should retain the original SS method as the more robust approach or at least compare the SS methods' influence.

2. In Figure S5, the error bars of the means are too large to justify a linear regression. This casts some doubt on the reliability of implementing this scheme into a global model. The large uncertainties may also indicate that the asymmetric effect is influenced by additional, unaccounted-for factors, and that the net contribution could remain negligible. Moreover, I think that the regression line in Figure S5 was derived only from the binned averages. This should be done using all dataset for U₁₀

between 5-20 m/s. There were only about 20 measurements over 20 m/s.

3. For Δs , the parameterization implies that the bubble effect at 20 m/s ($\Delta s = 0.077$) is only ~40% higher than at 5 m/s ($\Delta s = 0.053$). Although the authors set $\Delta s = 0$ at $U_{10} = 0$, extrapolating the linear fit back to zero wind speed still yields $\Delta s = 0.0045$. This formulation is not physically consistent, as both breaking-wave frequency and whitecap coverage are known to increase nonlinearly with wind speed (typically $\propto U_{10}^3$). Moreover, inert gas constraints (e.g., Stanley et al. 2009; Nicholson et al. 2011) support a highly increased bubble contribution at high winds and negligible at low winds. Consequently, the present parameterization likely overestimates bubble effects at low wind speeds while underestimating their importance under strong wind conditions. Overall, near-zero wind speeds should imply negligible bubble activity, and the abrupt jump in Δs around 5 m/s raises questions about whether the parameterization is physically justified.

4. Similar to Point 3, the current Δs model is not physically well established. A more defensible approach would be to regress the flux with separate interfacial and bubble terms that have distinct wind dependencies. Now the scheme may mis-attribute part of the interfacial flux to the bubble component, especially at moderate winds where bubble signals are weak. Δs is derived iteratively from those two equations, any measurement noise, bias in ΔC and misrepresentation in K_{int} can be redistributed into the Δs term, leading to potential overestimation of bubble effects.

Version 2:

Reviewer comments:

Reviewer #3

(Remarks to the Author)

Review #3 Dong Y., M. Yang, T. G. Bell, C. A. Marandino, D. K. Woolf: Asymmetric bubble-mediated gas transfer enhances global ocean CO₂ uptake, submitted to Nature Communications (2025)

The authors have responded my remarks in an appropriate manner. I found their response to Reviewer 4 concerns are convincing. This led to an improvement of the article by reinforcing the physical soundness of the analytical approach and the statistical robustness of the results. The manuscript is clear and well written.

Here are some typos that should be corrected:

- Equations (5) and (6) (and R6 and R7) should read

$$\Delta_s = 0.0132(1 - 1.37U_{10}^{-0.37})$$

$$\Delta_s = 0.0132(1 - 2.95U_{10}^{-0.67})$$

as in Figure S5 caption.

- Line 425 « represent »

Reviewer #4

(Remarks to the Author)

The authors have presented a clear and thorough explanation of their methodology and provided comprehensive responses demonstrating the effectiveness of their approach. I find the revisions satisfactory and the responses well-answered. The manuscript has been significantly improved in clarity, structure, and scientific rigor. I believe the revised version meets the standards for publication.

We thank the three reviewers for the time and effort they have taken to review our manuscript thoroughly. We have found the feedback very helpful, and we hope that you will find the revised manuscript much improved. We respond to individual comments and suggestions in detail below, and the resulting changes are tracked in the accompanying documents. The reviewer comments are given in **black**, our responses are given in **blue**, and the updated text is given in **orange**.

Please note that our response document is long because it contains repetitive answers to the three reviewers, where appropriate, rather than citing previous answers.

Reviewer #1

The authors use eddy covariance data to discuss the importance of an asymmetric contribution in air-sea CO₂ fluxes. Eddy covariance data which provide the flux and the partial pressure difference are used to infer an asymmetric parameterization. The authors argue that accounting for such asymmetric effect improve the match to the data. They then use the new parameterization to estimate global CO₂ fluxes using other global observation based partial pressure difference products, and find a significant different with numbers currently in the literature. In my opinion, the paper presents significant issues in terms of presentation, methodology and the subsequent interpretation.

Thank you for your time and feedback. We note your concerns with respect to presentation, methodology, and interpretation. Our response to specific points is detailed below.

Comment 1

The authors state that they use a “2D fit” to demonstrate the differences in evasion and invasion configurations, to demonstrate that the symmetric only formulation fails. A major problem to me is that the authors do not show their data in the main manuscript. They only show a resulting fit to the data in figure 1A which they use as proof of their argument. This is not acceptable as the reader should see the empirical evidence from their argument and assess whether the fit is improved when asymmetric contribution is considered. The data are shown in supplementary and it is not very convincing. When I look at figure S4, it is very difficult for me to get convinced that your symmetric fit and asymmetric fit are different in a statistical way. There is so much noise around the fitted line. Similarly, figure 1B is not convincing if you do not show the data.

Thank you for this important comment. We realize that our explanation in the original manuscript may not have been sufficiently clear. The key evidence supporting our argument is not in Figure 1A or 1B alone, but in the comparison between Figures 1A and 1B. Figure 1A shows that the symmetric formulation results in a visible divergence in the gas transfer velocity (K) between weak invasion, strong invasion, strong evasion, and weak evasion. Figure 1B demonstrates that the asymmetric formulation reduces this divergence, effectively collapsing the data toward a more consistent fit. This comparison highlights the improvement achieved by incorporating the asymmetric effect. To clearly explain this point, we added the following content in the main text near Fig. 1:

It is important to emphasize that the improvement offered by the asymmetric equation is not primarily demonstrated through a better statistical fit to noisy field data, but rather through the reduction in the systematic divergence between the four K^{2D}_{Sy} groups (as evidenced by the difference between Fig. 1A and Fig. 1B). The fit of K^{2D}_{Asy} is slightly lower than the fit of K^{2D}_{Sy} due to the inclusion of Δ_s (Table S1). Unlike the 1D fit, the 2D fit approach minimises misfit in flux rather than transfer velocity and enables inclusion of data for both weak evasion and weak invasion. Fitting K^{2D}_{Asy} to all EC data (~ 0.7) yields a substantially greater R^2 than the K^{1D}_{Asy} fit to the high flux data (~ 0.6 , Table S1). The robustness of the 2D method is evaluated in detail in the Materials and Methods.

Fig. S6 (formerly Fig. S4) in the supplement is intended to illustrate how the individual K values change when moving from the symmetric to the asymmetric parameterization derived from the 1D fit, specifically for the strong invasion and strong evasion cases. It is not meant to demonstrate the statistical improvement of the fit itself. All data used in this study will be shared publicly. We have extended the presentation of the individual data in the supplementary document (Figs. S3 and S7). First, the hourly eddy covariance flux observations are plotted against the wind speed and color-coded by ΔfCO_2 , and for comparison, the bulk flux calculated from the 2D fit and the asymmetric equation is also plotted on this figure. Second, we performed a statistical analysis of the agreement between the bulk flux estimates and the EC flux. The bulk flux estimated using the asymmetric equation and the 2D fit method agrees better with the observed EC CO_2 flux compared to the bulk flux estimated using the conventional symmetric equation and the 1D fit method (Fig. S7 and Table R1-1). This indicates that the asymmetric equation is more appropriate for the bulk CO_2 flux estimates.

Fig. S3. Illustration of the 2D fit: hourly sea-air CO_2 flux observations and estimates versus U_{10} . The dots represent the EC sea-air CO_2 flux observations, color-coded by the $\Delta f\text{CO}_2$. The crosses denote the bulk flux estimates using the asymmetric bulk equation (i.e., Equation 2 in the main text) and the 2D fit method.

Fig. S7. Comparison of EC flux observations with bulk flux estimates. Red squares show fluxes estimated using the symmetric equation and K parameterisation from the 1D fit (red line in Fig. S6); orange

squares and line indicate bin averages and the corresponding linear fit. The root mean square error (RMSE) and R^2 for the symmetric equation are 6.23 and 0.69, respectively. Blue dots show fluxes estimated using the asymmetric equation and K parameterisation from the 2D fit (lines in Fig. 1B); green squares and line indicate bin averages and the corresponding linear fit. The RMSE and R^2 for the asymmetric equation are 6.08 and 0.71, respectively. It shows that the asymmetric formulation aided by the 2D fit yields higher R^2 , lower RMSE, and is closer to the 1:1 line.

Table R1-1. The statistical metrics of the comparison between the bulk flux estimates and the EC flux observations.

		RMSE	R^2	$Flux_{EC} = a \times Flux_{bulk} + b$	
				a	b
Weak invasion	Symmetric & 1D	5.36	0.25	1.05	-0.22
	Asymmetric & 2D	5.34	0.25	1.01	0.09
Strong invasion	Symmetric & 1D	6.95	0.59	0.84	-1.48
	Asymmetric & 2D	6.76	0.62	0.92	-1.56
Strong evasion	Symmetric & 1D	4.16	0.38	0.74	0.68
	Asymmetric & 2D	3.94	0.41	0.96	0.31
Weak evasion	Symmetric & 1D	4.88	-0.04	0.43	0.56
	Asymmetric & 2D	4.73	0.03	0.76	0.56
All data	Symmetric & 1D	6.32	0.68	0.81	-1.91
	Asymmetric & 2D	6.13	0.70	0.94	-0.88

Given the large random uncertainty in individual data points (~30% on average), judging “by eye” the statistical significance of large data sets is not practical, and direct formal analyses of the original data often have weak statistical power. We have implemented a fair and transparent statistical method that is applied to composites of data (four data groups). The composite analysis has far greater statistical power and is effective in drawing out characteristics that cannot be observed in the individual data points.

Comparison of Fig. 1A and 1B is an effective presentation of the success of the asymmetric equation in “collapsing” the data. However, following all comments, we acknowledge that a thorough statistical analysis is necessary to confirm the separation of groups in Fig. 1A, and the improvement achieved with the asymmetric equation in Fig. 1B.

We added 95% confidence intervals (CIs) to different fits (see the revised Fig. 1 below). For the **symmetric equation**, the fits begin to diverge at wind speeds above 10 m s^{-1} . The T-test confirms that these differences are statistically significant (p -value < 0.05 , Fig. S4). The **asymmetric equation** does not exhibit statistically significant differences between groups (i.e., the fits largely converge), as shown by the p -values. These statistical tests support our findings that the asymmetric equation is superior.

Fig. 1. Gas transfer velocity (K_{660}) parameterisations with 10-meter neutral wind speed (U_{10}). Parameterisation of gas transfer velocity derived from hourly eddy covariance (EC) sea-air CO_2 observations and the 2D method with: (A) the symmetric bulk equation (Equation 1, K^{2D}_{sy}); and (B) the asymmetric bulk equation (Equation 2, K^{2D}_{Asy}). Red-solid lines, Weak Invasion ($-20 < \Delta f\text{CO}_2 \leq 0 \text{ } \mu\text{atm}$, mean = $-11 \text{ } \mu\text{atm}$, $R^2 = 0.22$, $N = 617$ hrs). Blue-dashed lines, Strong Invasion ($\Delta f\text{CO}_2 \leq -20 \text{ } \mu\text{atm}$, mean = $-68 \text{ } \mu\text{atm}$, $R^2 = 0.63$, $N = 2889$ hrs). Purple-dot-dashed lines, Strong Evasion ($\Delta f\text{CO}_2 \geq 20 \text{ } \mu\text{atm}$, mean = $29 \text{ } \mu\text{atm}$, $R^2 = 0.41$, $N = 236$ hrs). Green-dot lines, Weak Evasion ($0 < \Delta f\text{CO}_2 < 20 \text{ } \mu\text{atm}$, mean = $9 \text{ } \mu\text{atm}$, $R^2 = 0.014$, $N = 340$ hrs). Here, the R^2 refer to the fits in panel A; those for panel B are similar (see Table S1). The 95% confidence intervals (CI) are added to each parameterisation curve, using corresponding colours.

Fig. S4. T-test results (p -values) comparing different flux regime fits under the symmetric (A) and asymmetric (B) equations across a range of wind speeds. Red line: difference between weak invasion and strong invasion fits; Blue line: difference between strong invasion and strong evasion fits; Purple line: difference between strong evasion and weak evasion fits. Below the grey dashed line is significant ($p = 0.05$ threshold).

We also conducted a robustness test using a bootstrapping test (a specific kind of Monte Carlo method) by systematically removing one or more cruises from the dataset and analyzing the stability of the 2D collapse method. The results (now included in a new subsection "Robustness test of the 2D method" in the Materials and Methods) demonstrate that the asymmetric formulation consistently reduces the divergence in K estimates across different subsets, supporting the plausibility of our Δ_s assumption.

Robustness test of the 2D fitting method

To assess the robustness of the 2D fitting approach, we performed several sensitivity tests.

First, we applied a bootstrapping test by systematically excluding one or more cruises from the entire dataset (Fig. S11). The divergence between invasion and evasion groups observed in the symmetric equation consistently merges (Fig. S11A), and the asymmetric equation continues to reduce this divergence across all subsets (Fig. S11B). This consistency demonstrates that the observed pattern is not driven by a few specific cruises or one specific research group, and supports the stability of the asymmetric equation and the 2D fitting method. We note that the weak evasion

is always an outlier due to less data and high uncertainty, as shown in Fig. 1 and discussed in the main text.

Fig. S11A. Parameterisations of gas transfer velocity as a function of U_{10} , excluding one or more cruises at a time, using the symmetric equation. The cruises ANDREXII, JR18007, AMT28, AMT29, JR18004, JR19001, JR19002, and JR30001 were all conducted by the Plymouth Marine Laboratory (PML) aboard two UK research vessels using consistent instrumentation and processing protocols (Dong et al., 2021a).

Fig. S11B. Parameterisations of gas transfer velocity as a function of U_{10} , excluding one or more cruises at a time, using the asymmetric equation.

Comment 2

Line 158: the authors state: “but we have demonstrated that asymmetric bubble exchange is important for sea-air CO₂ transfer”. I am sorry but they have not demonstrated anything. They are stating something but the empirical evidence is not shown in a convincing way. We note the concern and as explained in our response to the previous comment, we have strengthened the arguments attached to our analysis. We have also acknowledged the contribution of previous studies (e.g., Stanley et al., 2022; Liang et al., 2013) that reached a similar conclusion from independent evidence. Some caution is appropriate, and while we find the evidence compelling, we acknowledge some room for uncertainty in the revised text.

Abstract: ...Further collection of evasion data is recommended to improve understanding of the asymmetric factor and reduce its uncertainty. Our study suggests that the global ocean may have taken up more CO₂ than previously thought, and the asymmetric bulk equation should be used for sea-air CO₂ flux estimates...

Introduction: ...The asymmetric transfer of highly insoluble gases such as noble gases, is well-evidenced by observations of their supersaturation state in the field (e.g., Stanley et al., 2009) and laboratory (e.g., Stanley et al., 2022). However, the saturation state cannot be used to evaluate the asymmetric transfer of CO₂ because of the effect of biological activity and seawater CO₂ buffering capacity. Zhang (2012) estimated that asymmetric bubble transfer accounts for more than 20% of the total oceanic CO₂ uptake based on a CO₂ supersaturation factor scaled from oxygen. Results for these very poorly soluble gases are useful to provide an upper limit, but extrapolations from other gases to infer asymmetric effects on CO₂ are likely to be unreliable...

Results (Section on Impact of asymmetry on large-scale CO₂ flux estimates): ...The GCB calculates sea-air CO₂ flux using the symmetric bulk equation, but previous results provide evidence of bubble-induced asymmetry in gas exchange (e.g., Stanley et al., 2009; Liang et al., 2013; Leighton et al., 2018), and our results further support that asymmetric bubble-mediated exchange is important for sea-air CO₂ transfer. Here we assess the impact of the asymmetric transfer on global sea-air CO₂ flux estimates...

Discussion and conclusions: ...The EC sea-air CO₂ flux dataset is dominated by measurements in net invasion conditions (86%, N = 3506 hrs), whereas there are fewer net evasion observations (N = 576 hrs), which limits our confidence in the global asymmetry-adjusted ocean CO₂ uptake estimate. Nevertheless, the value of Δ_s (~0.5% on average) used in this study is consistent with existing evidence. Stanley et al. (2009) reported Xenon (Xe) supersaturation of ~1% under typical ocean conditions. The solubility of Xe (α ~0.1 at 20 °C) is lower than that of CO₂ (α ~0.7 at 20 °C), meaning that the Δ_s of CO₂ is expected to be less than 1%. Another independent estimate uses a bubble dynamic model designed for low solubility gases, and extrapolates a ~0.7% supersaturation factor for CO₂ (Liang et al., 2013). Still, more direct sea-air CO₂ flux measurements are needed to reduce the uncertainty associated with the bubble-induced supersaturation factor, and strengthen and improve the asymmetric parameterisation (Equation 3) proposed here...

Comment 3

Next, I would focus on data at high wind speed, there are not much bubbles at 5 m/s so it is unclear why they are considered. I would use at least 10 or 15 m/s as cut-off. You will have less data but maybe better quality. Sensitivity to the amount of data used need to be discussed.

Thank you for raising this point. While bubble-mediated processes are generally stronger at higher wind speeds, previous studies have shown that wave breaking, and thus potential bubble generation, can begin at wind speeds as low as 3 m/s from both field observations (Hanson and Phillips, 1999; Monahan and O'Muircheartaigh, 1986) and satellite-based derivation of whitecap coverage (Salisbury et al., 2013). Further direct evidence for bubble involvement at these lower wind speeds comes from observed divergence between CO₂ and DMS transfer velocities starting near 5 m s⁻¹ (Fig. R1-1, left), suggesting that the bubble effect on CO₂ exchange starts to be detectable already at low-to-moderate wind speeds.

In addition, our goal is to develop a parameterization that applies across the full range of wind conditions for global scaling. Considering that moderate wind speeds (5-10 m/s) dominate the global wind speed distribution (see Fig. R1-1, right), it is appropriate to retain the full dataset rather than apply a higher wind speed cutoff.

[FIGURE REDACTED]

Fig. R1-1. Left: CO₂ and DMS observations in the field (Bell et al., 2013, 2017; Blomquist et al., 2017; Landwehr et al., 2018; Zavarsky et al., 2018). Right: Global ocean wind speed distribution from Wanninkhof (2014).

Comment 4

I would suggest plots similar to the one presented in Stanley et al 2022, where a supersaturation is evaluated, which the authors could estimate through a ratio of water and air side concentration or partial pressure. They could then demonstrate that for the same partial pressure difference, the flux is much larger in the case where bubbles do play a role (invasion) than in evasion configuration. The authors need to find a way to demonstrate the claim in their data, which is not done in the present version of the manuscript. I would insist that the work from Stanley (and others) clearly demonstrate the existing of the asymmetric term in the flux. Also, there are some existing asymmetric parameterization in the literature, Stanley et al 2009 is one; Liang et al 2013 is another; and while they were presented for other gases, the authors might be able to compare to their proposed version since they include some dependency in diffusivity and solubility.

We fully agree that the existence of an asymmetric term is well supported by previous studies, including field measurements (e.g., Emerson et al., 2019; Stanley et al., 2009), lab experiments (Stanley et al., 2022), and bubble model simulations (Leighton et al., 2018; Liang et al., 2013). These references were not sufficiently discussed in our original manuscript, and we have now added to the Introduction, Results and Discussion, and Conclusions sections (see our reply to **comment 2**). We now specifically discuss the asymmetric parameterizations proposed by Stanley et al. (2009) and Liang et al. (2013), which serve as useful upper bounds and support the plausibility of our estimated asymmetry factor.

Regarding the suggestion to apply the supersaturation-based method as in Stanley et al. (2022), we appreciate this idea, but there are several limitations in applying it to our dataset:

1) Small signal for CO₂: Bubble-induced supersaturation for CO₂ is typically only ~0.5–1% under normal wind conditions (<20 m/s), translating to a change in $\Delta f\text{CO}_2$ of just ~2–4 μatm . This signal is subtle compared to the natural variability and measurement uncertainty in field CO₂ flux observations.

2) Uneven date distribution: As noted in the manuscript, we have fewer evasion observations than invasion observations, which limits the direct comparison as suggested by the reviewer. However, our methodology allows us to evaluate the asymmetric effect indirectly through changes in K . For example, the asymmetry has a proportionally larger impact on weak invasion cases (e.g., 10–20% change at $\Delta f\text{CO}_2 = 20 \mu\text{atm}$) than the strong invasion cases (4-8% change at $\Delta f\text{CO}_2 = 50 \mu\text{atm}$), offering a way to constrain the effect despite limited evasion data.

Our method of analysis is effective for demonstrating the success of the asymmetric equation in better representing CO₂ flux data from the field. This study is the first that presents evidence from field CO₂ data. While we acknowledge previous model results for CO₂ and experimental evidence for the least soluble gases, this new line of evidence is important, since it is field-based and specific to CO₂ (the most climate-related gas) fluxes.

Comment 5

Moving to the global flux estimates, I think the discussion is questionable. Of course if you take the same delta p CO₂ data and change the kw formulation, you will get large differences. But what if you had implemented your asymmetric formulation in a global ocean model? Would the model equilibrate to similar flux after some transient? This has been the argument of the modeling community to keep using the Wanninkhof quadratic fit.

We appreciate this important point. First, we would like to clarify that our global flux estimates are still based on the Wanninkhof (2014) parameterization, as stated in line 345 of the manuscript. Therefore, the flux differences shown are not due to a change in the gas transfer velocity formulation. Instead, they arise from the inclusion of the bubble-induced supersaturation term within the asymmetric framework (Δ_s in Equation 2, main paper). To emphasise this, we further added the following information in the section “Impact of asymmetry on large-scale CO₂ flux estimates”:

To ensure comparability, all flux estimates use the same $K_{660-U_{10}}$ parameterisation from Wanninkhof (2014), with coefficients scaled to the ERA5 wind speed (Hersbach et al., 2020).

As for the asymmetry fit in the data-based flux estimates and in the models, in the original manuscript, we had already: (1) Discussed the two types of estimate: observation-based bulk flux estimates and GOBM output; (2) Calculated via modified bulk formulae that apply directly and completely to the observation-based bulk flux estimate. We acknowledge that the effect of any surface formulations is reduced in GOBM model estimates (e.g., Bellenger et al., 2023), and have already discussed this in the context of the skin effect (Table 1). In this section, we focus on the observation-based bulk flux estimate and show that this estimate of the uptake is affected when considering asymmetry. We welcome working with global modelers to improve their representations of sea-air interactions in global ocean biogeochemistry models.

Reviewer #2

April 3, 2025

Review of: Asymmetric bubble-mediated gas transfer enhances global ocean CO₂ uptake, by Dong et al.,

This paper argues that the very impressive results of air-sea CO₂ exchange fluxes from Eddy Correlation (EC) measurements at sea indicate that there is evidence for an asymmetric exchange flux of CO₂ at the air-water interface due to the enhanced invasion rate from bubble fluxes created by wave breaking. Background arguments in the “Materials and Methods” and in the “Supplemental Information” present a strong case that the CO₂ EC flux method has resulted in gas exchange coefficient, k_{660} , values as a function of wind speed that are broadly consistent with values determined by other methods based on bomb ¹⁴C and gas tracer release experiments. Furthermore, the great difference between k_{660} determined with CO₂ and DMS eddy correlation experiments suggest that bubble exchange plays a significant role in CO₂ air-sea exchange. This is a meaningful review of the existing data that have been published (e.g., Yang et al., 2022; Leighton et al., 2018). The point of this revisit to the CO₂ EC results is to make the case that evidence for a bubble effect on the CO₂ gas exchange rate exists in the compiled data from field experiments. The authors make the argument that CO₂ EC data cannot be successfully interpreted as entirely due to symmetric gas exchange in which global air-sea transport by CO₂ invasion and evasion are caused by the same mechanism. Rather, they suggest a model in which there is a small excess CO₂ invasion flux that is due to the transport by bubbles. It is well known that the bubble fluxes during air-sea exchange of the highly insoluble atmospheric gases of N₂, O₂, and Ar cause measurable supersaturation in surface waters. CO₂, however, is much more soluble than these gases and the role of bubbles is more difficult to evaluate. The authors argue that one has to use a model that accounts for asymmetric air-sea exchange to interpret the existing data. The asymmetric factor is the supersaturation, Δs , caused by bubble fluxes. If Δs is zero the bubble flux is zero, and as you increase Δs the evidence for the importance of bubbles increases to the point where it makes a significant difference in the calculation of the amount of anthropogenic CO₂ that has entered the ocean, which the authors do convincingly.

Thank you for your thoughtful and comprehensive summary of our work, and for recognizing the significance of our study.

Comment 1

I am impressed by this paper: the authors have written it well, and I believe they have done an impressive job of interpreting the marine CO₂ EC data, which is a challenge because there is a lot of scatter. However, I do not think they have made a strong case, that, as stated in the abstract and conclusions, “This study provides direct field evidence of asymmetric bubble-mediated transfer...” They show that evidence for the importance of bubble-mediated CO₂ exchange in their analysis depends strongly on the value of Δ_s , and it is pretty clear that they are uncertain about what this value is. I think this is a major flaw in the paper and makes one skeptical of the conclusions about the anthropogenic fluxes. I would recommend that, at the very least, the authors do an error analysis (something like a Monte Carlo uncertainty calculation) to determine how well one needs to know Δ_s to be able to constrain the importance of bubbles in the interpretation of air-sea CO₂ exchange.

We appreciate your recognition of our efforts and share your concern regarding the uncertainty in quantifying the asymmetric factor, Δ_s . Our study provides field-based evidence suggesting the importance of asymmetric bubble-mediated CO₂ exchange, but we agree that the value of Δ_s is still uncertain. In response, we have taken several steps to address this limitation, following your recommendation:

1) We added 95% confidence intervals (CIs) to different fits (see the revised Fig. 1 below). For the **symmetric equation**, the fits begin to diverge at wind speeds above 10 m s⁻¹. The T-test confirms that these differences are statistically significant (p -value < 0.05, Fig. S4). **The asymmetric equation** does not exhibit statistically significant differences between groups (i.e., the fits largely converge), with the exception of the weak evasion group, as shown by the p -values. These statistical tests support our findings that the asymmetric equation is superior.

Fig. 1. Gas transfer velocity (K_{660}) parameterisations with 10-meter neutral wind speed (U_{10}). Parameterisation of gas transfer velocity derived from hourly eddy covariance (EC) sea-air CO_2 observations and the 2D method with: **(A)** the symmetric bulk equation (Equation 1, K^{2D}_{sy}); and **(B)** the asymmetric bulk equation (Equation 2, K^{2D}_{Asy}). Red-solid lines, Weak Invasion ($-20 < \Delta f\text{CO}_2 \leq 0 \mu\text{atm}$, mean = $-11 \mu\text{atm}$, $R^2 = 0.22$, $N = 617$ hrs). Blue-dashed lines, Strong Invasion ($\Delta f\text{CO}_2 \leq -20 \mu\text{atm}$, mean = $-68 \mu\text{atm}$, $R^2 = 0.63$, $N = 2889$ hrs). Purple-dot-dashed lines, Strong Evasion ($\Delta f\text{CO}_2 \geq 20 \mu\text{atm}$, mean = $29 \mu\text{atm}$, $R^2 = 0.41$, $N = 236$ hrs). Green-dot lines, Weak Evasion ($0 < \Delta f\text{CO}_2 < 20 \mu\text{atm}$, mean = $9 \mu\text{atm}$, $R^2 = 0.014$, $N = 340$ hrs). Here, the R^2 refer to the fits in panel A; those for panel B are similar (see Table S1). The 95% confidence intervals (CI) are added to each parameterisation curve, using corresponding colours.

Fig. S4. *T*-test results (*p*-values) comparing different flux regime fits under the symmetric (A) and asymmetric (B) equations across a range of wind speeds. Red line: difference between weak invasion and strong invasion fits; Blue line: difference between strong invasion and strong evasion fits; Purple line: difference between strong evasion and weak evasion fits. Below the grey dashed line is significant ($p = 0.05$ threshold).

2) Empirical uncertainty analysis. We also conducted a robustness test using a bootstrapping test (a specific kind of Monte Carlo method) by systematically removing one or more cruises from the dataset and analyzing the stability of the 2D collapse method. The results (now included in a new subsection "Robustness test of the 2D method" in the Materials and Methods) demonstrate that the asymmetric formulation consistently reduces the divergence in K estimates across different subsets, supporting the plausibility of our Δs assumption.

Robustness test of the 2D fitting method

To assess the robustness of the 2D fitting approach, we performed several sensitivity tests.

First, we applied a bootstrapping test by systematically excluding one or more cruises from the entire dataset (Fig. S11). The divergence between invasion and evasion groups observed in the symmetric equation consistently merges (Fig. S11A), and the asymmetric equation continues to reduce this divergence across all subsets (Fig. S11B). This consistency demonstrates that the observed pattern is not driven by a few specific cruises or one specific research group, and supports the stability of the asymmetric equation and the 2D fitting method. We note that the weak evasion is always an outlier due to less data and high uncertainty, as shown in Fig. 1 and discussed in the main text.

Fig. S11A. Parameterisations of gas transfer velocity as a function of U_{10} , excluding one or more cruises at a time, using the symmetric equation. The cruises ANDREXII, JR18007, AMT28, AMT29, JR18004, JR19001, JR19002, and JR30001 were all conducted by the Plymouth Marine Laboratory (PML) aboard two UK research vessels using consistent instrumentation and processing protocols (Dong et al., 2021a).

Fig. S11B. Parameterisations of gas transfer velocity as a function of U_{10} , excluding one or more cruises at a time, using the asymmetric equation.

3) Literature-based justification for Δ_s used in this study. We added the following paragraph to the Discussion and Conclusions: ...The EC sea-air CO_2 flux dataset is dominated by measurements in net invasion conditions (86%, $N = 3506$ hrs), whereas there are fewer net evasion observations ($N = 576$ hrs), which limits our confidence in the global asymmetry-adjusted ocean CO_2 uptake estimate. Nevertheless, the value of Δ_s ($\sim 0.5\%$ on average) used in this study is consistent with existing evidence. Stanley et al. (2009) reported Xenon (Xe) supersaturation of $\sim 1\%$ under typical ocean conditions. The solubility of Xe ($\alpha \sim 0.1$ at 20°C) is lower than that of CO_2 ($\alpha \sim 0.7$ at 20°C), meaning that the Δ_s of CO_2 is expected to be less than 1% . Another independent estimate uses a bubble dynamic model designed for low solubility gases, and extrapolates a $\sim 0.7\%$ supersaturation factor for CO_2 (Liang et al., 2013). Still, more direct sea-air CO_2 flux measurements are needed to

reduce the uncertainty associated with the bubble-induced supersaturation factor, and strengthen and improve the asymmetric parameterisation (Equation 3) proposed here ...

4) Revision of the wording in the Abstract. We have rephrased our claims to clearly reflect the existing uncertainty in Δ_s .

...Further collection of evasion data is recommended to improve understanding of the asymmetric factor and reduce its uncertainty. Our study suggests that the global ocean may have taken up more CO₂ than previously thought, and the asymmetric bulk equation should be used for sea-air CO₂ flux estimates...

Reviewer #3

Review #1 of Dong Y., M. Yang, T. G. Bell, C. A. Marandino, D. K. Woolf: Asymmetric bubble-mediated gas transfer enhances global ocean CO₂ uptake submitted to Nature Communications (2025)

The article is well written and is addressing very important and long-lasting issue on the global impact of bubble-mediated CO₂ air-sea exchanges. To do so, the authors propose an original approach based on the re-analysis of a large amount of in situ Eddy Covariance (EC) flux measurements. The authors showed that assuming an asymmetric equation for the CO₂ flux better explain the in-situ measurements. New values for the parameters are determined to provide an estimate of the global impact of bubble-mediated CO₂ transfer. The impact of this process is to increase the global ocean CO₂ sink by about 25%. The authors also re-assess the impact of the ocean cool skin on CO₂ exchanges that is only applicable to the interfacial part of the flux and not to the bubble-mediated one. Finally, the authors discuss the fact that taking into account this process increases the disagreement between model-based and observational product-based assessments of the global ocean CO₂ sink. Since, ocean is a major compartment of the terrestrial Carbon cycle, this study is therefore likely to be of interest to a wide audience. The approach seems appropriate to back the results although **some details on their statistical robustness are lacking** (see below). Multiple references to the “Methods” and “Supplementary material” sections make it hard to follow. Working on the structure of these annexes and pointing specifically to sub-sections in the main text could help the reader.

We sincerely thank the reviewer for the positive and constructive comments. We agree that the manuscript would benefit from a more dedicated uncertainty analysis and that referencing to content in the “Materials and Methods” and “Supplementary Material” sections should be more specific. The revised text specifies subsections. In response, all co-authors have worked together to improve the clarity and readability of the manuscript.

Analytical approach:

Comment 1

Measurements from many campaigns are used in this study. This means that a diversity of sensors, a diversity of post-treatments of these data and on the top of this a diversity of physical conditions are used in this analysis. In addition, all these differences are not equally represented in the total

ensemble and may not be independent: For example, all the weak evasion situations might all be taken from a single campaign and suffer from a common experimental bias. How is this taken into account?

This is a good point. Indeed, using data from multiple campaigns introduces variability due to differences in instrumentation, data processing, and physical conditions. We have taken several steps to minimize these effects and to ensure that our ensemble analysis remains robust:

1) Strict data quality selection: We included only high-quality, modern EC datasets (see the criteria in Yang et al., (2022)). Datasets with known issues, such as unreliable instrumentation or outdated processing, were excluded from the ensemble. This helps reduce potential biases introduced by inconsistent methodology.

2) Consistent data processing framework: Half of the datasets (from different groups) were reprocessed using a unified correction approach, as detailed in Yang et al. (2022). For new datasets not covered in that synthesis, we applied the same correction procedures, ensuring methodological consistency across the full dataset (Dong et al., 2021a, 2024; Ford et al., 2024).

3) Expanded documentation and added a new figure: We added a plot of time series of EC fluxes and $\Delta f\text{CO}_2$ for all cruises in the Supplementary Material (Figs. S1B, S1C). This makes it easier to assess the spread and context of the evasion data.

Additionally, the bootstrapping test (see our reply to your comment 3) also indicates robustness of the EC data and no evident inherent bias due to a specific dataset.

Fig. S1. Eddy covariance (EC) data collected from 17 cruises. ...B and C: Time series of EC flux and $\Delta f\text{CO}_2$, respectively, from multiple cruises, with each cruise separated by vertical dashed lines. Cruise names are labeled above the bottom axis of panel C. The SOAP, ANDREXII, NBP (NBP-1210 and NBP-1402), HiWinGS, Knorr07 (Knorr07a and Knorr07b), Knorr11, and SO (SO-234 and SO-235) correspond to the 11 synthesized cruises (i.e., red lines in panel A). AMT28 and AMT29 represent the two cruises in the Atlantic Ocean (i.e., yellow lines in panel A), while the remaining four cruises are additional collections in the Southern Ocean (i.e., blue lines in panel A).

Comment 2

The authors separate the EC data by invasion/evasion conditions but the information of the amount of data for each condition and on their origin among the 17 campaigns are lacking. This could be achieved by presenting the K – U10 scatterplots for the 4 conditions that are defined in this study or providing histograms of number of measurements as a function of U10 and delta_fCO2.

Thank you for the helpful suggestion. In the caption of Fig. 1, we included the number of data points in each group, but we agree that this may not be sufficient. To address this, we now provide a histogram showing the number of measurements as a function of wind speed (U_{10}) and $\Delta f\text{CO}_2$,

which has been added to the Supplementary Material (Figs. S2B and S2C). We believe that the newly added histograms, together with the time series plots (Figs. S1B, S1C), offer sufficient information about the distribution and origin of the data under different conditions.

Regarding the suggestion to include $K-U_{10}$ scatterplots: this is not feasible for the weak invasion and weak evasion conditions due to the issue of dividing by ΔC that is close to zero (see Methods and Materials). This is precisely why we adopted the 2D regression method, which fits the fluxes directly as a function of both U_{10} and $\Delta f\text{CO}_2$. As an alternative to show all data, we added the EC flux vs U_{10} plot and color-coded it by $\Delta f\text{CO}_2$ and the comparison between the EC flux and the bulk flux with symmetric and asymmetric equations in the Supplementary Material (Figs. S3, S7)

Fig. S2. Distributions of the data used in this study. ...B and C: Histograms of the U_{10} (B) and $\Delta f\text{CO}_2$ (C) for the data used in this study.

Fig. S3. Illustration of the 2D fit: hourly sea-air CO_2 flux observations and estimates versus U_{10} . The dots represent the EC sea-air CO_2 flux observations, color-coded by the $\Delta f\text{CO}_2$. The crosses denote the bulk flux estimates using the asymmetric bulk equation (i.e., Equation 2 in the main text) and the 2D fit method.

Fig. S7. Comparison of EC flux observations with bulk flux estimates. Red squares show fluxes estimated using the symmetric equation and K parameterisation from the 1D fit (red line in Fig. S6); orange squares and line indicate bin averages and the corresponding linear fit. The root mean square error (RMSE)

and R^2 for the symmetric equation are 6.23 and 0.69, respectively. Blue dots show fluxes estimated using the asymmetric equation and K parameterisation from the 2D fit (lines in Fig. 1B); green squares and line indicate bin averages and the corresponding linear fit. The RMSE and R^2 for the asymmetric equation are 6.08 and 0.71, respectively. It shows that the asymmetric formulation aided by the 2D fit yields higher R^2 , lower RMSE, and is closer to the 1:1 line.

Comment 3

Quality of the 2D fit should be characterized. In addition, robustness of the difference between symmetric and asymmetric models for K should be tested. This could be done with some kind of bootstrapping method, for example, by rejecting one or few campaigns data at a time.

We appreciate your suggestion. This point was also raised by Reviewer 2. In response, we have taken several steps to address this limitation, following your recommendation:

1) Add 95% confidence intervals (CIs) onto the fits in Fig. 1 to show if the difference is statistically significant. We added 95% confidence intervals (CIs) to different fits (see the revised Fig. 1 below). For the **symmetric equation**, the fits begin to diverge at wind speeds above 10 m s^{-1} . The T-test confirms that these differences are statistically significant (p -value < 0.05 , Fig. S4). **The asymmetric equation** does not exhibit statistically significant differences between groups (i.e., the fits largely converge), with the exception of the weak evasion group, as shown by the p -values. These statistical tests support our findings that the asymmetric equation is superior.

Fig. 1. Gas transfer velocity (K_{660}) parameterisations with 10-meter neutral wind speed (U_{10}). Parameterisation of gas transfer velocity derived from hourly eddy covariance (EC) sea-air CO_2 observations and the 2D method with: (A) the symmetric bulk equation (Equation 1, K^{2D}_{Sy}); and (B) the asymmetric bulk equation (Equation 2, K^{2D}_{Asy}). Red-solid lines, Weak Invasion ($-20 < \Delta f\text{CO}_2 \leq 0 \mu\text{atm}$, mean = $-11 \mu\text{atm}$, $R^2 = 0.22$, $N = 617$ hrs). Blue-dashed lines, Strong Invasion ($\Delta f\text{CO}_2 \leq -20 \mu\text{atm}$, mean = $-68 \mu\text{atm}$, $R^2 = 0.63$, $N = 2889$ hrs). Purple-dot-dashed lines, Strong Evasion ($\Delta f\text{CO}_2 \geq 20 \mu\text{atm}$, mean = $29 \mu\text{atm}$, $R^2 = 0.41$, $N = 236$ hrs). Green-dot lines, Weak Evasion ($0 < \Delta f\text{CO}_2 < 20 \mu\text{atm}$, mean = $9 \mu\text{atm}$, $R^2 = 0.014$, $N = 340$ hrs). Here, the R^2 refer to the fits in panel A; those for panel B are similar (see Table S1). The 95% confidence intervals (CI) are added to each parameterisation curve, using corresponding colours.

Fig. S4. T-test results (p -values) comparing different flux regime fits under the symmetric (A) and asymmetric (B) equations across a range of wind speeds. Red line: difference between weak invasion and strong invasion fits; Blue line: difference between strong invasion and strong evasion fits; Purple line: difference between strong evasion and weak evasion fits. Below the grey dashed line is significant ($p = 0.05$ threshold).

2) Empirical uncertainty analysis. We also conducted a robustness test using a bootstrapping test (a specific kind of Monte Carlo method) by systematically removing one or more cruises from the dataset and analyzing the stability of the 2D collapse method. The results (now included in a new subsection "Robustness test of the 2D method" in the Materials and Methods) demonstrate that the asymmetric formulation consistently reduces the divergence in K estimates across different subsets, supporting the plausibility of our Δs assumption.

Robustness test of the 2D fitting method

To assess the robustness of the 2D fitting approach, we performed several sensitivity tests.

First, we applied a bootstrapping test by systematically excluding one or more cruises from the entire dataset (Fig. S11). The divergence between invasion and evasion groups observed in the symmetric equation consistently merges (Fig. S11A), and the asymmetric equation continues to reduce this divergence across all subsets (Fig. S11B). This consistency demonstrates that the observed pattern is not driven by a few specific cruises or one specific research group, and supports the stability of the asymmetric equation and the 2D fitting method. We note that the weak evasion is always an outlier due to less data and high uncertainty as shown in Fig. 1 and discussed in the main text.

Fig. S11A. Parameterisations of gas transfer velocity as a function of U_{10} , excluding one or more cruises at a time, using the symmetric equation. The cruises ANDREXII, JR18007, AMT28, AMT29, JR18004, JR19001, JR19002, and JR30001 were all conducted by the Plymouth Marine Laboratory (PML)

aboard two UK research vessels using consistent instrumentation and processing protocols (Dong et al., 2021a).

Fig. S11B. Parameterisations of gas transfer velocity as a function of U_{10} , excluding one or more cruises at a time, using the asymmetric equation.

3) Literature-based justification for Δ_s used in this study. We added the following paragraph to the Discussion and Conclusions: ...The EC sea-air CO₂ flux dataset is dominated by measurements in net invasion conditions (86%, N = 3506 hrs), whereas there are fewer net evasion observations (N = 576 hrs), which limits our confidence in the global asymmetry-adjusted ocean CO₂ uptake estimate. Nevertheless, the value of Δ_s (~0.5% on average) used in this study is consistent with existing evidence. Stanley et al. (2009) reported Xenon (Xe) supersaturation of ~1% under typical ocean conditions. The solubility of Xe (α ~0.1 at 20 °C) is lower than that of CO₂ (α ~0.7 at 20 °C),

meaning that the Δ_s of CO₂ is expected to be less than 1%. Another independent estimate uses a bubble dynamic model designed for low solubility gases, and extrapolates a ~0.7% supersaturation factor for CO₂ (Liang et al., 2013). Still, more direct sea-air CO₂ flux measurements are needed to reduce the uncertainty associated with the bubble-induced supersaturation factor, and strengthen and improve the asymmetric parameterisation (Equation 3) proposed here ...

4) Revision of the wording in the Abstract. Although we have done the above analysis, the large uncertainty in the Δ_s value is still hard to quantify. Therefore, we have rephrased our claims to clearly reflect the existing uncertainty.

...Further collection of evasion data is recommended to improve understanding of the asymmetric factor and reduce its uncertainty. Our study suggests that the global ocean may have taken up more CO₂ than previously thought, and the asymmetric bulk equation should be used for sea-air CO₂ flux estimates...

Clarity and context

The article is clear and well written. Here are some suggestions to improve the presentation of the context:

Comment 4

In addition to bubbles and cool skin that are discussed here, other processes may be sources of asymmetries in the air-sea CO₂ transfer: Sea sprays which occur in the same conditions as bubbles and rainfall that tends to increase the ocean CO₂ uptake. The authors should discuss these processes for their results based on the recent literature.

We agree that additional processes beyond bubbles and the cool skin effect may contribute to asymmetries in sea-air CO₂ exchange, and we now include a discussion of these processes in the uncertainty analysis section of the main text. The revised text is as follows:

Second, the results shown in Fig. 1 do not consider the cool skin effect, which can also lead to bias in the derived K_{660} . However, the cool skin effect is relatively more substantial at low wind speeds and relatively weak and consistent at intermediate and high wind speeds (Donlon et al., 2002). Inclusion of the cool skin effect in the derivation of K^{2D} does not collapse any divergences shown in Fig. 1A (see Fig. S12). Other processes, such as sea spray and rainfall (both of which often occur during stormy conditions), may also induce asymmetric gas transfer. Sea spray tends to enhance evasion relative to invasion (Staniec et al., 2021), which is opposite to the pattern observed

in Fig. 1. Sea spray-related asymmetries remain highly uncertain (Woolf, 2025) and are beyond the scope of this study. Rain events may also promote asymmetric CO₂ uptake (Parc et al., 2024; Ho & Schanze, 2020), but rain intensity is generally independent of wind speed and thus unlikely to explain the wind speed-dependent divergence observed in Fig. 1.

Some suggestion to improve the clarity of the article:

Comment 5

There are too many notations for the transfer velocity (sometimes not defined as $K_{CO_2_true}$), they should be homogenized maybe with symmetric/asymmetric in subscript, obs/1D/2D in superscript and drop the references to the Schmidt number value and to CO₂ as they do not bare any useful information.

Thank you. According to your comments, we have revised the $K_{660_CO_2}$ to K^{1D_asy} , K^{1D_sy} , K^{2D_asy} , and K^{2D_sy} accordingly in all text and figures. For example, the previous text in caption Fig. 1:

“Parameterisation of $K_{660_CO_2}$ derived from eddy covariance (EC) sea-air CO₂ observations with: (A) the symmetric bulk equation (Equation 1); and (B) the asymmetric bulk equation (Equation 2).”

Revised text: Parameterisation of gas transfer velocity derived from hourly eddy covariance (EC) sea-air CO₂ observations and the 2D method with: (A) the symmetric bulk equation (Equation 1, K^{2D_sy}); and (B) the asymmetric bulk equation (Equation 2, K^{2D_Asy}).

Comment 6

As presented in the “Materials and Methods” section, the link between the asymmetry and the over-pressure factors is not clear.

We agree that their relationship is not clearly explained. We should mention that the relation $\Delta_s = \delta K_{bub}/(K_{bub} + K_{int})$ is derived from equations 2 and 4. Below is the derivation process:

$$Flux = K[C_w - C_a(1 + \Delta_s)] \quad (2)$$

$$Flux = K_{int}(C_w - C_a) + K_{bub}[C_w - C_a(1 + \delta)] \quad (4)$$

Replace K in Equation 2 as $K_{int} + K_{bub}$, and combine Equations 2 and 4:

$$(K_{int} + K_{bub})[C_w - C_a(1 + \Delta_s)] = K_{int}(C_w - C_a) + K_{bub}[C_w - C_a(1 + \delta)]$$

Cancel out the same term and rearrange this equation, we can get:

$$\Delta_s = \delta K_{bub}/(K_{int} + K_{bub})$$

We revised the text “ Δ_s can be approximated from δ using $\Delta_s = \delta K_{bub}/(K_{bub} + K_{int})$ ” as “Combining Equations 2 and 4, Δ_s can be derived from δ as $\Delta_s = \delta K_{bub}/(K_{bub} + K_{int})$ ”.

Minor suggestions

Comment 7

Line 53: can you explain why bubble mediated transfer dependence on solubility is due to bubble finite volume?

Sure. Solubility dependence arises from the change in content (or partial pressure) of each gas within a bubble while submerged due to exchange across the surface of the bubble. If the bubble had (effectively) infinite volume, no amount of transfer would change the partial pressure. For a finite volume, more soluble gases rapidly equilibrate with the surrounding water, while the equilibration is slower for less soluble gases. Equilibration suppresses further net transfer, resulting in the solubility dependence.

For illustration, consider a bubble with a lifetime of 3 s in the water. A highly soluble gas may equilibrate with the surrounding seawater within ~ 1 s, leaving only that first second for efficient exchange, while a poorly soluble gas may remain far from equilibrium and continue exchanging throughout the entire 3 s. In this simple case, the bubble-mediated transfer velocity for the poorly soluble gas would be roughly three times that of the highly soluble gas. In reality, the situation is more complex. We added a reference here to guide readers who wish to explore the topic in more detail:

...depends on solubility because bubbles have limited volume and lifetime (Woolf, 1993);

Comment 8

Lines 100-104, maybe the inclusion of equation (S4) could help the discussion here.

Good point. We refer to Equation S4 here.

From theory, this bias is expected to be largest when $\Delta f\text{CO}_2$ is small and wind speed is high, when bubble-mediated transfer is enhanced (see Supplementary Material, Section 1, Equation S4).

Comment 9

Can the statistical significance in the different gas transfer velocities obtained for different situations be tested? Can this information be added in Figure 1?

Yes, we have added the 95% confidence interval to Figure 1. See our reply to your comment 3.

Comment 10

Supplementary Material line 61, what is the reference for the K_{int} parametric expression that is used here?

It is Blomquist et al. (2017), which is indicated in line 53. But we also added it in line 61:

- $K_{int} = (Sc/660)^{-1/2} (0.74U_{10}^{1.33})$ (Blomquist et al., 2017).

Comment 11

The reference to Supp. Mat. should precise the number of the section where details are provided.

Thank you. We added the section number in the Supplement Material to the main text.

- when bubble-mediated transfer is enhanced (see Supplementary Material, Section 1, Equation S4).
- The fit is to the flux, meaning that the error minimisation is on the predicted flux (i.e., a ‘least squares’ fit to flux; see Supplementary Material, Section 2).
- a δ of 0.0132 (Leighton et al., 2018) and a K_{int} parameterisation based on EC DMS observations (Blomquist et al., 2017; Fig. S8) are used to reanalyse the EC CO₂ flux data (Supplementary Material, Section 3).

Comment 12

EC fluxes re-analysis and fitting of K seems to be made on hourly means but this is never clearly stated.

Yes, thank you for pointing this out. We added “hourly” to the caption of Fig. 1:

Parameterisation of gas transfer velocity derived from hourly eddy covariance (EC) sea-air CO₂ observations and the 2D method

Comment 13

Lines-139-141: Why is a 1D fit used here instead of the 2D fit?

This is a good point. In the case that the asymmetric equation is used for analyzing the high flux data (i.e., with $|\Delta fCO_2|$ restricted to $> 20 \mu\text{atm}$), both the 1D and 2D methods are suitable to use and their results are very similar (see Table S1). In fact, the 2D fit has a better R^2 compared to the 1D fit. To make the results consistent, we use the 2D fit to all data in this paragraph and in the supporting Fig. S8:

The $K_{2D_Asy}^{2D}$ based on all EC data is consistent with the $K_{660-U_{10}}$ parameterisation that has been constrained by the global ¹⁴C inventory (Wanninkhof, 2014; Fig. S8).

Fig. S8. Parameterisations of K_{660} - U_{10} based on different gas observations. Red line: the 2D fit to all the EC sea-air CO_2 flux data used in this study, with the red shadow representing the 95% confidence interval ($K_{660}^{\text{2D}} = 0.48U_{10}^{1.75}$, $R^2 = 0.71$). Blue-dot-dashed line: K_{660} based on global ^{14}C inventory (Wanninkhof, 2014). Yellow-dashed line: $K_{660,\text{DMS}}$ based on EC sea-air DMS flux observations (Blomquist et al., 2017), a proxy for interfacial gas transfer that assumes minimal bubble-mediated transfer of DMS

Comment 14

Lines 161 and 238-240: The authors should clearly provide the final expression for K_{asy} (and Δ_s) that is recommended as equation (3).

Good point. We now present the Δ_s expression as equations 3 and 4.

$$\Delta_s = 0.00016U_{10} + 0.0045 \quad U_{10} \geq 5 \text{ m s}^{-1} \quad (3)$$

The specific form of the K parameterisation is not central to this study, and the upscaling analysis uses the Wanninkhof (2014) parameterisation for consistency with previous work. Therefore, it is not included in the main text but is clearly presented in Fig. S8 and Table S1.

Comment 15

Lines 197-200: It has been shown that depressions in the Southern Ocean can induce outgassing due to decreased atmospheric pressure and entrainment of DIC-rich waters at the base of the mixed layer. This is not taken into account here and could reduce the impact of asymmetric transfer. Please comment.

This is an important and interesting finding (e.g., Nicholson et al., 2022; Toolsee et al., 2024). However, any changes in atmospheric pressure or entrainment of DIC-rich waters should already be reflected in reliable atmospheric and surface seawater $f\text{CO}_2$ observations. If SOCAT data or biogeochemical models fail to capture this effect, the resulting flux estimates will be biased. Here, we assume the GCB flux estimate represents the best available knowledge to date and use it as a baseline to investigate a new effect—bubble-mediated gas exchange. This does not imply the flux estimate is perfect; other processes, such as the depression effect, should be considered separately.

Reference

- Bell, T. G., Landwehr, S., Miller, S. D., De Bruyn, W. J., Callaghan, A. H., Scanlon, B., et al. (2017). Estimation of bubble-mediated air-sea gas exchange from concurrent DMS and CO₂ transfer velocities at intermediate-high wind speeds. *Atmospheric Chemistry and Physics*. <https://doi.org/10.5194/acp-17-9019-2017>
- Bell, T. G., De Bruyn, W., Marandino, C. A., Miller, S. D., Law, C. S., Smith, M. J., & Saltzman, E. S. (2015). Dimethylsulfide gas transfer coefficients from algal blooms in the Southern Ocean. *Atmospheric Chemistry and Physics*, *15*(4), 1783–1794. <https://doi.org/10.5194/acp-15-1783-2015>
- Bellenger, H., Bopp, L., Ethé, C., Ho, D., Duvel, J., Flavoni, S., et al. (2023). Sensitivity of the global ocean carbon sink to the ocean skin in a climate model. *Journal of Geophysical Research: Oceans*, *128*. <https://doi.org/10.1029/2022JC019479>
- Blomquist, B. W., Brumer, S. E., Fairall, C. W., Huebert, B. J., Zappa, C. J., Brooks, I. M., et al. (2017). Wind speed and sea state dependencies of air-sea gas transfer: Results from the High Wind Speed Gas Exchange Study (HiWinGS). *Journal of Geophysical Research: Oceans*, *122*(10), 8034–8062. <https://doi.org/10.1002/2017JC013181>
- Butterworth, B. J., & Miller, S. D. (2016). Air-sea exchange of carbon dioxide in the Southern Ocean and Antarctic marginal ice zone. *Geophysical Research Letters*, *43*(13), 7223–7230. <https://doi.org/10.1002/2016GL069581>
- Dong, Y., Yang, M., Bakker, D. C. E., Kitidis, V., & Bell, T. G. (2021a). Uncertainties in eddy covariance air-sea CO₂ flux measurements and implications for gas transfer velocity parameterisations. *Atmospheric Chemistry and Physics*, *21*(10), 8089–8110. <https://doi.org/10.5194/acp-21-8089-2021>
- Donlon, C. J., Minnett, P. J., Gentemann, C., Nightingale, T. J., Barton, I. J., Ward, B., & Murray, M. J. (2002). Toward improved validation of satellite sea surface skin temperature measurements for climate research. *Journal of Climate*, *15*(4), 353–369. [https://doi.org/10.1175/1520-0442\(2002\)015<0353:TIVOSS>2.0.CO;2](https://doi.org/10.1175/1520-0442(2002)015<0353:TIVOSS>2.0.CO;2)
- Emerson, S., Yang, B., White, M., & Cronin, M. (2019). Air-Sea Gas Transfer: Determining

- Bubble Fluxes With In Situ N₂ Observations. *Journal of Geophysical Research: Oceans*, 124(4), 2716–2727. <https://doi.org/10.1029/2018JC014786>
- Ford, D. J., Shutler, J. D., Blanco-sacristán, J., Corrigan, S., Bell, T. G., Yang, M., et al. (2024). Enhanced ocean CO₂ uptake due to near-surface temperature gradients. *Nature Geoscience*, 1-6. <https://doi.org/10.1038/s41561-024-01570-7>
- Hanson, J. L., & Phillips, O. M. (1999). Wind sea growth and dissipation in the open ocean. *Journal of Physical Oceanography*, 29(8 PART 1), 1633–1648. [https://doi.org/10.1175/1520-0485\(1999\)029<1633:wsgadi>2.0.co;2](https://doi.org/10.1175/1520-0485(1999)029<1633:wsgadi>2.0.co;2)
- Hersbach, H., Bell, B., Berrisford, P., Hirahara, S., Horányi, A., Muñoz-Sabater, J., et al. (2020). The ERA5 global reanalysis. *Quarterly Journal of the Royal Meteorological Society*, 146(730), 1999–2049. <https://doi.org/10.1002/qj.3803>
- Ho, D. T., & Schanze, J. J. (2020). Precipitation - induced reduction in surface ocean pCO₂: Observations from the eastern tropical Pacific Ocean. *Geophysical Research Letters*, 47(15), e2020GL088252. <https://doi.org/10.1029/2020GL088252>
- Leighton, T. G., Coles, D. G. H., Srokosz, M., White, P. R., & Woolf, D. K. (2018). Asymmetric transfer of CO₂ across a broken sea surface. *Scientific Reports*, 8(1), 1–9. <https://doi.org/10.1038/s41598-018-25818-6>
- Liang, J. H., Deutsch, C., McWilliams, J. C., Baschek, B., Sullivan, P. P., & Chiba, D. (2013). Parameterizing bubble-mediated air-sea gas exchange and its effect on ocean ventilation. *Global Biogeochemical Cycles*, 27(3), 894–905. <https://doi.org/10.1002/gbc.20080>
- Monahan, E. C., & O’MUIRCHEARTAIGH, I. G. (1986). Whitecaps and the passive remote sensing of the ocean surface. *International Journal of Remote Sensing*, 7(5), 627–642. <https://doi.org/10.1080/01431168608954716>
- Nicholson, S. A., Whitt, D. B., Fer, I., du Plessis, M. D., Lebéhot, A. D., Swart, S., et al. (2022). Storms drive outgassing of CO₂ in the subpolar Southern Ocean. *Nature Communications*, 13(1), 1–12. <https://doi.org/10.1038/s41467-021-27780-w>
- Parc, L., Bellenger, H., Bopp, L., Perrot, X., & Ho, D. T. (2024). Global ocean carbon uptake enhanced by rainfall. *Nature Geoscience*, 17(9), 851–857. <https://doi.org/10.1038/s41561->

024-01517-y

- Salisbury, D. J., Anguelova, M. D., & Brooks, I. M. (2013). On the variability of whitecap fraction using satellite-based observations. *Journal of Geophysical Research: Oceans*, *118*(11), 6201–6222. <https://doi.org/10.1002/2013JC008797>
- Stanley, R. H. R., Jenkins, W. J., Lott, D. E., & Doney, S. C. (2009). Noble gas constraints on air-sea gas exchange and bubble fluxes. *Journal of Geophysical Research: Oceans*, *114*(11), 1–14. <https://doi.org/10.1029/2009JC005396>
- Stanley, R. H. R., Kinjo, L., Smith, A. W., Aldrett, D., Alt, H., Kopp, E., et al. (2022). Gas fluxes and steady state saturation anomalies at very high wind speeds. *Journal of Geophysical Research: Oceans*, *127*(10), 1–19. <https://doi.org/10.1029/2021jc018387>
- Staniec, A., Vlahos, P., & Monahan, E. C. (2021). The role of sea spray in atmosphere–ocean gas exchange. *Nature Geoscience*, *14*(8), 593–598. <https://doi.org/10.1038/s41561-021-00796-z>
- Toolsee, T., Nicholson, S. A., & Monteiro, P. M. S. (2024). Storm - Driven pCO₂ Feedback Weakens the Response of Air - Sea CO₂ Fluxes in the Sub - Antarctic Southern Ocean. <https://doi.org/10.1029/2023GL107804>
- Wanninkhof, R. (2014). Relationship between wind speed and gas exchange over the ocean revisited. *Limnology and Oceanography: Methods*, *12*(6), 351–362. <https://doi.org/10.4319/lom.2014.12.351>
- Woolf, D. (2025). Eight Categories of Air–Water Gas Transfer. *Oceans*, *6*(2), 27. <https://doi.org/10.3390/oceans6020027>
- Yang, M., Bell, T. G., Bidlot, J. R., Blomquist, B. W., Butterworth, B. J., Dong, Y., et al. (2022). Global synthesis of air-sea CO₂ transfer velocity estimates from ship-based eddy covariance measurements. *Frontiers in Marine Science*, *9*(6), 1–15. <https://doi.org/10.3389/fmars.2022.826421>
- Zavarsky, A., Goddijn-Murphy, L., Steinhoff, T., & Marandino, C. A. (2018). Bubble-mediated gas transfer and gas transfer suppression of DMS and CO₂. *Journal of Geophysical Research: Atmospheres*, *123*(12), 6624–6647. <https://doi.org/10.1029/2017JD028071>

Zhang, X. (2012). Contribution to the global air-sea CO₂ exchange budget from asymmetric bubble-mediated gas transfer. *Tellus, Series B: Chemical and Physical Meteorology*, 64(1).
<https://doi.org/10.3402/tellusb.v64i0.17260>

We thank the two reviewers for further valuable comments, which provides an additional
opportunity for us to improve the manuscript. We respond to individual comments and suggestions
in detail below, and the resulting changes are tracked in the accompanying documents. The
reviewer comments are given in **black**, our responses are given in **blue**, and the updated text is
given in **orange**.

**Reviewer #3:**

Dong Y., M. Yang, T. G. Bell, C. A. Marandino, D. K. Woolf: Asymmetric bubble-mediated gas
transfer enhances global ocean CO₂ uptake, submitted to Nature Communications (2025)

Using a large collection of in situ eddy covariance measurements, the authors show that assuming
an asymmetric model to take into account the impact of bubbles-mediated CO₂ exchanges better
explain the in-situ measurements. This is evident as the transfer velocity coefficient obtained from
an asymmetric formulation is less dependent on the CO₂ concentration difference across the air-
sea interface. The new parameterization is used to calculate the global ocean CO₂ sink that is about
25% larger than previous estimates. The authors also re-assess the impact of the ocean cool skin
on CO₂ exchanges that is only applicable to the interfacial part of the flux and not to the bubble-
mediated one.

The first version of the manuscript was suffering from (1) a lack of details on the representativity
of the data and a lack of statistical robustness assessment of the fits and (2) a lot of different
notations, a complex structure and a lack of reference to specific subsections of supplementary
material. Overall, the authors addressed these two main drawbacks in a convincing way. They in
particular provide a lot more details on the data and on the statistical significance of the change
when considering an asymmetric formulation. Details are given below.

**(1) Analytical approach**

- In order to show that the dataset used encompasses a large variety of situations, and that each
condition is documented by independent cruises, the authors now provide an overview of the data
obtained in different conditions of CO₂ gradient and wind speed (Figure S1 and S2). It has yet to
be noted that, on Figure S2, it is hard to clearly see the relative sampling of the different categories

of conditions (weak/strong invasion/evasion and wind speed). A way could be to provide a 2D
 histogram (number of hours as a function of delta fCO₂ and U₁₀) or to break the histogram bars
 of Fig S2B for the different weak/strong invasion/evasion cases.

- The new version of Figure 1 provides new information on the uncertainty behind the fit. A
 statistical test is now performed (Figure S4) that shows that the use of an asymmetric formulation
 makes the transfer coefficients for different conditions (i.e. weak/strong evasion/invasion)
 statistically indistinguishable, as it should be.

- The robustness of the findings is further tested by performing a bootstrap test consisting in
 excluding cruises from the complete dataset. The authors show that the same conclusions can be
 drawn for each case (Figure S11). This result could maybe be made more quantitative and synthetic
 by defining a metric of the spread of the transfer coefficients functions for each panel of Fig S11
 and then compare statistically the two ensembles shown on S11A and S11B.

Thank you very much for acknowledging the improvements made to our manuscript in response
 to the reviewers' comments. We appreciate your further suggestions and have made the following
 revisions accordingly:

- • Update the Fig S2.

**Fig. S2. Distributions of the data used in this study.** **A:** Mean $\Delta f\text{CO}_2$ versus wind speed (U_{10}) for four
groups of data, with an averaging bin size of 2 m s^{-1} wind speed. Four data categories are shown: weak
invasion ($-20 < \Delta f\text{CO}_2 \leq 0 \text{ } \mu\text{atm}$, red dots), strong invasion ($\Delta f\text{CO}_2 \leq -20 \text{ } \mu\text{atm}$, red squares), strong evasion
($\Delta f\text{CO}_2 \geq 20 \text{ } \mu\text{atm}$, blue stars), and weak evasion ($0 < \Delta f\text{CO}_2 < 20 \text{ } \mu\text{atm}$, blue diamonds). **B:** Histograms of
the $\Delta f\text{CO}_2$ for the data used in this study. **C-F:** Histograms of the U_{10} for the four data categories (defined
the same as in panel A).

- • We appreciate the reviewer's suggestion to include a metric to illustrate the spread of the k
values in Fig. 1 and Fig. S11. While we acknowledge the importance of quantifying data, we
found it challenging to identify a specific metric that would effectively convey this information
without potentially complicating the interpretation. We believe that the spread among the four
lines is already visually evident in Fig. 1A and Fig. S11, and our goal is to maintain clarity in
our presentation. We hope this explanation is satisfactory.

(2) Clarity of the manuscript

The new manuscript is clear and the supplementary material subsections are referenced clearly in
the text. Yet, I still have some comments that could help make the manuscript easier to follow:

- Some supplementary figures are not or hardly commented: Fig S1B and S1C, S3.

Thank you. We added new text in the main text to discuss these figures:

In lines 101-102: Each scenario includes data collected from multiple cruises (Figs. S1B and S1C).

In lines 122-124: The bulk flux derived from the 2D fitting approach generally replicates the hourly
EC flux observations across various conditions (Fig. S3).

- Table S1 that has many panels that are hardly exploited, are they all necessary?

Table S1 is intended to provide comprehensive information regarding the comparison between 2D
and 1D fits, as well as symmetric and asymmetric equations. This table serves as a valuable
resource for readers who wish to delve deeper into the fitting details. As it is included in the
supplementary material (not in the main text), we would prefer to retain all the information in
Table S1.

Minor comments

The authors finally considered the minor comments I made on the previous version and adapted
the manuscript accordingly.

- I would like to suggest to the authors to provide the line numbers in their reply so that reviewers
can efficiently track the changes.

This is a good suggestion. We added line number in this reply document and will also take this
suggestion in the future. Thank you.

- Line 118: “K1D_Asy” not “_Sy”

We have corrected the typo in the caption of Fig. S6 in the supplementary material. Thank you for
bringing this to our attention.

**Reviewer #4:**

Review of “Asymmetric bubble-1 mediated gas transfer enhances global ocean CO₂
Uptake” by Dong et al.

Summarization:

This study investigates bubble-mediated air–sea CO₂ exchange using eddy-covariance
observations from 17 historical cruises. The authors develop an asymmetric parameterization
scheme for CO₂ flux, demonstrating that bubble processes enhance oceanic uptake. Applying this
scheme at the global scale, they estimate that the ocean absorbs an additional ~0.5 Pg C yr⁻¹ (over
25% increase) compared with conventional symmetric formulations, highlighting the substantial
impact of bubble asymmetry on the global carbon budget.

I mainly have some concerns about implementing asymmetric equations in global ocean models,
given the limitations of both the methodology and data.

We sincerely thank the reviewer for their fair and constructive comments. We acknowledge that
the manuscript still has weaknesses regarding the treatment of Δ_s . In response, all co-authors have
collaboratively worked to address your concerns and enhance the clarity of the manuscript.

Our results are presented in terms of two distinct overpressure factors (Δ_s and δ) and we begin our
detailed response by giving a full explanation of each. The δ (overpressure factor or oversaturation
factor) is directly linked with the bubble term and has a more intuitive interpretation: it represents
the overpressure of the gas within the bubble primarily due to hydrostatic pressure. In contrast, Δ_s
represents an effective oversaturation factor: the oversaturation induced by the bubble which is
partially offset by interfacial transfer processes. The distinction between these two parameters can
be better understood through the following equations:

$$108 \quad Flux = K[C_w - C_a(1 + \Delta_s)]$$

$$109 \quad Flux = K_{int}(C_w - C_a) + K_{bub}[C_w - C_a(1 + \delta)]$$

Here, Δ_s in the first equation represents a mixed effect of interfacial and bubble processes, while δ
in the second Equation specifically corresponds to the pure bubble effect. By combining these
equations, we can relate Δ_s and δ as follows:

$$113 \quad \Delta_s = \delta K_{bub} / (K_{bub} + K_{int})$$

This relationship indicates that Δ_s represents δ multiplied by the fractional contribution of the
bubble-mediated gas transfer velocity to the total gas transfer velocity. This distinction is crucial
for understanding how these factors influence gas exchange dynamics. Their relationship has been
explained in the Materials and Methods section (subsection ‘Re-analysis of EC CO₂ fluxes
assuming asymmetric transfer’).

To improve the clarity of the main text, we move the equation in the Materials and Methods to the
introduction and further explain the difference and linkage between Δ_s and δ .

**Lines 75-88: If the overall gas transfer (K) is mechanistically separated into the interfacial transfer**
**component (K_{int}) and the bubble-mediated transfer component (K_{bub}) (Woolf, 1993), Equation 2**
**can be expressed as:**

$$124 \quad Flux = K_{int}(C_w - C_a) + K_{bub}[C_w - C_a(1 + \delta)] \quad (3)$$

The first term in the right side of Equation 3 represents the interfacial transfer process, which is
symmetric, whereas the second term corresponds to the bubble-mediated transfer process, which
is asymmetric (represented by the over-pressure factor, δ). Note that δ and Δ_s have different

meanings: δ is only related to the bubble process, while Δ_s captures the combined effects of both
bubble and interfacial processes. By combining Equations 2 and 3, Δ_s and δ can be related as:

$$\Delta_s = \delta K_{bub} / (K_{int} + K_{bub}) \quad (4)$$

Field observations, such as the supersaturation of noble gases (e.g., Stanley et al., 2009), typically
reflect Δ_s , since the natural measurements integrate both interfacial and bubble processes. δ can be
simulated by bubble dynamic models based on the near-surface bubble size distributions (e.g.,
Leighton et al., 2018).

**Major Comment:**

**1. The effectiveness of this 2D asymmetric method is substantiated by the reduced divergence**
of K under weak evasion, strong invasion, and strong evasion conditions. However, this alone is
not convincing evidence, as incorporating U_{10} linearly into Δ_s then asymmetric equation will,
by design, reduce the disparity in Figure1(a) by extra polynomial terms. Meanwhile, the new
weighting method indeed cuts off the influence of low ΔC cases in another way and could
artificially overestimate K or Δ_s with flux/wind speed increased. In my view, the authors should
retain the original SS method as the more robust approach or at least compare the SS methods'
influence.

Thank you for your comments, which provide us with an opportunity to further improve the
robustness of the 2D method.

First, we would like to emphasize that the inclusion of the Δ_s term is supported by the theory that
bubble-mediated gas exchange can produce overpressure, leading to asymmetry. The
quantification of Δ_s is based on an independent estimation of the overpressure factor (δ) and the
interfacial transfer velocity (K_{int}). The δ is estimated using a bubble dynamic model that
incorporates near-surface bubble size distribution observations (Leighton et al., 2018), while K_{int}
is derived from eddy covariance DMS observations (Blomquist et al., 2017). Therefore, the
inclusion and quantification of Δ_s are supported by scientific evidence and were not arbitrarily
designed to meet our objectives.

However, we agree with the reviewer that the linear relationship between Δ_s and U_{10} was not
treated with sufficient rigor, both mathematically and physically. In response, we have first revised
our mathematical treatment of Δ_s . We would like to clarify that the original sum of squares (SS)

approach (i.e., the 1D method) relies on the derivation of K from EC flux observations, which can
be unreliable under weak invasion/evasion conditions due to significant uncertainties in the EC
flux and $\Delta f\text{CO}_2$, as discussed in the manuscript. In contrast, the 2D fitting method directly relates
flux to U_{10} and ΔC in the following manner: $\text{EC flux} = \Delta C_{660}(aU_{10}b + c)$. This approach eliminates
the need for K derivation and allows us to include weak invasion/evasion data in our analysis.
While the 2D method does relatively less weight on the low ΔC data within each group, it does
not imply that K or Δ_s is artificially overestimated with increasing flux or wind speed. As shown
in Fig. S2A, the mean $\Delta f\text{CO}_2$ (and thus ΔC) remains consistent across different wind speeds for all
scenarios, except in strong invasion cases. This indicates that the data are equally weighted across
varying wind speeds in the new SS method. For the strong invasion cases, the traditional SS method
(1D method) yields a function of $K_{660} = 0.44U_{10}^{1.84}$, while the new SS method (2D method)
produces $K_{660} = 0.45U_{10}^{1.80}$. The results from both parameterizations for strong invasion fall
between the weak invasion and strong evasion curves, indicating that they do not significantly
differ from one another.

We include some information in the Supplementary Material Section 2: The theory of 2D fit.

Lines 41-47: Additionally, Fig. S2A indicates that the mean $\Delta f\text{CO}_2$ (and consequently ΔC_{660})
remains relatively consistent across different wind speeds for each group, except in the case of
strong invasion scenarios. This suggests that the 2D fit applies equal weighting under varying wind
conditions, similar to the 1D fit. For the strong invasion scenario, the results of the 2D fit closely
align with those of the 1D fit, with both approaches indicating lower K values compared to weak
invasion scenarios and higher values compared to the strong evasion cases (Table S1).

**2. In Figure S5, the error bars of the means are too large to justify a linear regression.** This
casts some doubt on the reliability of implementing this scheme into a global model. The large
uncertainties may also indicate that the asymmetric effect is influenced by additional,
unaccounted-for factors, and that the net contribution could remain negligible. Moreover, I think
that the regression line in Figure S5 was derived only from the binned averages. This should be
done using all dataset for U_{10} between 5-20 m/s. There were only about 20 measurements over
20 m/s.

We acknowledge that the uncertainty associated with the derived Δ_s is substantial. This is primarily
due to the inherent uncertainties in the eddy covariance (EC) data and the sensitivity of Δ_s to these

uncertainties. The error bars shown in the original manuscript are ± 1 standard deviation.
Uncertainty would be based on standard error since the number of data points should be weighted
for the uncertainty; we updated the figure (See the updated Fig. S5, or the figure in the reply to
comment 4). Second, we do not think the asymmetric contribution is negligible. Because
“negligible” would require a zero or tiny Δ_s , and while the fitting method can be debated, a
substantial (0.5-1%) asymmetry cannot really be disputed. This Δ_s can significantly influence air-
sea CO₂ flux estimates.

In response to the reviewer’s suggestion, we agree that there are only a few data points above 20
195 m s⁻¹ wind speed, and the fit based on data for U_{10} between 5-20 m s⁻¹ would be more appropriate.
However, we still think fitting the bin averages should be better than fitting the hourly data point,
because the data is not distributed equally at different wind speeds (see Fig. S2). Fitting the hourly
dataset will overweight the moderate wind speed conditions (8-12 m s⁻¹) since most of the data are
concentrated in this moderate wind speed range, but under-weight the lower and higher wind
conditions. So, we will update the parameterisation by fitting the bin averages between 5-20 m s⁻¹.
However, as indicated by the reviewer’s next two comments, the linear wind speed dependence
of Δ_s is implausible physically. We invite the reviewer to refer to our responses to your next two
comments for details of the new parameterisations and the corresponding revision of the
manuscript.

**3. For Δ_s ,** the parameterization implies that the bubble effect at 20 m/s ($\Delta_s = 0.077$) is only ~40%
higher than at 5 m/s ($\Delta_s = 0.053$). Although the authors set $\Delta_s = 0$ at $U_{10} = 0$, extrapolating the
linear fit back to zero wind speed still yields $\Delta_s = 0.0045$. This formulation is not physically
consistent, as both breaking-wave frequency and whitecap coverage are known to increase
nonlinearly with wind speed (typically $\propto U_{10}^3$). Moreover, inert gas constraints (e.g., Stanley et
al. 2009; Nicholson et al. 2011) support a highly increased bubble contribution at high winds and
negligible at low winds. Consequently, the present parameterization likely overestimates bubble
effects at low wind speeds while underestimating their importance under strong wind conditions.
Overall, near-zero wind speeds should imply negligible bubble activity, and the abrupt jump in Δ_s
around 5 m/s raises questions about whether the parameterization is physically justified.

**4. Similar to Point 3, the current Δ_s model is not physically well established.** A more defensible
approach would be to regress the flux with separate interfacial and bubble terms that have distinct

wind dependencies. Now the scheme may mis-attribute part of the interfacial flux to the bubble
component, especially at moderate winds where bubble signals are weak. Δ_s is derived iteratively
from those two equations, any measurement noise, bias in ΔC and misrepresentation in K_{int} can
be redistributed into the Δ_s term, leading to potential overestimation of bubble effects.

Comments 3 and 4 concern the same topic, and we therefore respond to them together below.

- • **Δ_s estimation.** We agree with other studies that the contribution of bubbles to transfer velocity
is highly non-linear with respect to wind speed. But this refers to the bubble-mediated gas
transfer velocity (K_{bub}). The sensitivity of Δ_s may be less pronounced than one might intuitively
expect. As previously explained, $\Delta_s = \delta K_{bub} / (K_{int} + K_{bub})$ or $\Delta_s = \delta(K - K_{int}) / K$. This
means that the value of Δ_s is "emergent", relying on the assumptions made regarding δ , K_{int} ,
and K .
- • **Δ_s sensitivity to δ .** The sensitivity of Δ_s to wind speed would increase if δ were more sensitive
to changes in wind speed. For moderately soluble gases like CO_2 , the primary contribution of
bubbles to gas exchange arises from larger bubbles, which means δ for CO_2 is predominantly
driven by hydrostatic pressure associated with the effective bubble penetration depth. We
believe that this penetration depth does not significantly vary with wind speed, as supported
by theoretical argument (Keeling, 1993) and recent field observations (Czerski et al., 2022).
Additionally, for CO_2 , a gas is continually and rapidly adjusting to the hydrostatic pressure
until the bubble surfaces; therefore, having been much deeper early in its journey is almost
irrelevant (e.g., Woolf and Thorpe, 1991), differing from the much less soluble noble gases.
Therefore, we have fixed δ at 0.0132 according to the independent estimate from Leighton et
al. (2018). It is important to note that with this fixed value, Δ_s will asymptotically approach the
value of δ (i.e., 1.32%), reflecting a gradual increase that is reasonable for CO_2 . In contrast,
less soluble gases may exhibit different behaviors due to the influence of deeper bubbles.
- • **Δ_s sensitivity to K_{int} .** The sensitivity of Δ_s to the assumed interfacial component (K_{int}) is
noteworthy, as this could lead to a significant exaggeration of effects near 5 m s^{-1} if we have
slightly underestimated the direct contribution and, consequently, overestimated the bubble
contribution. The K_{int} we employed is derived from the EC DMS observations:

$$245 \quad K_{660int} = 0.74U_{10}^{1.33} \quad (\text{R1})$$

Given that the solubility of DMS is considerably higher than that of CO₂, the bubble
 contribution is expected to be minimal, allowing K_{int} to effectively represent the interfacial
 contribution. Although the parameterization for K_{int} here is based on a single cruise (Blomquist
 et al., 2017), it is consistent with the mean state across the existing 12 EC DMS cruises (see
 Fig. 5 in Dong et al., 2025).

- • **Δ_s sensitivity to K .** Δ_s is also sensitive to the total gas transfer velocity (K). Given that K_{int} has
 been fixed using an EC DMS-based parameterisation, Δ_s will vary depending on the chosen K .
 This sensitivity is particularly strong under low wind speeds (i.e., $\sim 5 \text{ m s}^{-1}$). In the manuscript,
 we use the EC air-sea CO₂ flux observations to represent the total gas exchange and apply an
 iterative method to derive Δ_s . However, these EC air-sea CO₂ fluxes inherently include the
 effect of chemical enhancement (CE), which is absent in EC DMS observations. While CE is
 negligible at high wind speeds, it becomes relatively important under low wind conditions.
 Given that Δ_s is sensitive to the treatment of K at low wind speeds, we should have removed
 the CE contribution from the observed EC CO₂ transfer velocities. Otherwise, the CE signal
 could be misinterpreted as a bubble-mediated effect, leading to an overestimation of Δ_s . To
 account for this, we applied a CE correction ratio derived from analytical and numerical models
 (Fairall et al., 2022; Luhar et al., 2018), defined and parameterised as:

$$263 \quad f_{CE} = K_{CE}/K - 1 = 18.12U_{10}^{-2.37} \quad (R2)$$

Accordingly, the CO₂ transfer velocity without chemical enhancement is given by $K_{no_CE} = K/$
 $(1 + f_{CE})$. This formulation assumes a carbonate reaction time constant of 4 s, resulting in an
 average chemical enhancement of $\sim 2.5 \text{ cm hr}^{-1}$ for CO₂, consistent with independent estimates
 (Wanninkhof et al., 2009; Wanninkhof & Knox, 1996). The Δ_s is not very sensitive to the
 uncertainty in f_{CE} at wind speeds higher than 5 m s^{-1} since the chemical enhancement is not
 significant under this wind condition.

- • **Δ_s parameterisation.** Based on the above treatments of δ , K_{int} , and K , we re-analysed the EC
 CO₂ data using the iterative method. The resulting Δ_s estimates are shown in the updated Fig.
 S5B (blue dots). Given that $\Delta_s = \delta(K - K_{int})/K$, and both K and K_{int} can be expressed as
 $a_1U_{10}^{b_1}$ and $a_2U_{10}^{b_2}$, respectively, Δ_s can thus be parameterised as $\Delta_s = \delta(1 - a_3/U_{10}^{b_3})$. Fitting
 the bin averages of Δ_s for U_{10} between $5\text{-}20 \text{ m s}^{-1}$ using the above form of the function, we
 obtain the following parameterisation:

$$276 \quad \Delta_s = 0.0132(1 - 1.37/U_{10}^{-0.37}) \quad (R3)$$

• **Alternative approach to estimate Δ_s .** Formulation R3 is based on the reanalysis of EC air-sea
CO₂ flux data to represent the K . Alternatively, Δ_s can be estimated using existing K
parameterisations:

$$280 K_{660} = 0.167U_{10}^2 + 1.203U_{10} + 0.36 \quad (\text{R4})$$

$$281 K_{660} = 0.25U_{10}^2 \quad (\text{R5})$$

Equation R4 is based on a synthesis of EC air-sea CO₂ flux data (Yang et al., 2022), whereas
the widely used formulation (Equation R5) is based on the ¹⁴C inventory (Wanninkhof, 2014).
It is important to note that Equation R4 includes the chemical enhancement and therefore
requires correction using Equation R2. In contrast, Equation R5 does not include CE, and the
impact of CE is attributed to the uncertainty by the author (Wanninkhof, 2014). Using
Equations R4 and R5 yields the following Δ_s parameterisations:

$$288 \Delta_s = 0.0132(1 - 1.65/U_{10}^{-0.44}) \quad (\text{R6})$$

$$289 \Delta_s = 0.0132(1 - 2.95/U_{10}^{-0.67}) \quad (\text{R7})$$

Formulations R3, R6, and R7 show general agreement. All exhibit a minor Δ_s values at very
low wind speeds ($< 5 \text{ m s}^{-1}$), thereby avoiding the abrupt jump around 5 m s^{-1} observed in
earlier formulations. They also show a gradual increase of Δ_s at higher wind speed,
asymptotically approaching the value of δ when the bubble-mediated exchange dominates the
total gas exchange. These characters are consistent with the physical considerations. At high
wind speeds ($10\text{-}20 \text{ m s}^{-1}$), Δ_s values from all three parameterisations agree within 10%.
However, at low wind speed (especially near 5 m s^{-1}), significant discrepancies arise due to the
strong sensitivity of Δ_s to the assumed K_{int} and K , highlighting large uncertainties in Δ_s under
such wind conditions. The Equations R5 and R6 are all based on EC air-sea CO₂ data, and
estimate similar Δ_s values (Figure S5B), therefore we only show Equation R5 in the main text.

 **Fig. S5. Gas transfer velocity and constrained asymmetric factor (Δ_s) versus U_{10} .** **A:** Interfacial transfer
 velocity (K_{int}) based on EC DMS observations (red-solid line; Blomquist et al., 2017) and total transfer
 velocity (K) from the EC air-sea CO₂ flux observations (Blue-solid line; Yang et al., 2022) and from the
 ¹⁴C inventory (green-dashed line; Wanninkhof, 2014). The parameterisation from Yang et al. (2022) with
 a chemical enhancement correction (i.e., $K/(1 + f_{CE})$; see the text in section 3) is shown as the purple-dashed
 line. **B:** Δ_s constrained using different approaches. Blue dots represent 1 m s⁻¹ binned averages of Δ_s derived
 from EC data reanalysis, with error bars indicating ± 1 standard error. The red line (Equation 5 in the main
 text, $R^2 = 0.11$) is a least-squares fit to the blue dots for wind speeds of 5-20 m s⁻¹. Data at $U_{10} < 5$ m s⁻¹ are
 excluded from the fit because bubble-mediated and thus asymmetric transfer is expected to be negligible.
 The green and purple dashed curves show the parameterised Δ_s based on the Wanninkhof (2014) and Yang
 et al. (2022, with chemical enhancement correction) parameterisations of K , respectively.

• **Treatment of Δ_s at U_{10} less than 5 m s⁻¹.** Comparative analysis of the EC CO₂ and DMS
 observations indicates that the CO₂ transfer velocity begins to increasingly exceed that of DMS
 at approximately 5 m s⁻¹ (see Figs. 5 and 6 in Dong et al., 2025). Consequently, we define 5 m
 s⁻¹ as the threshold wind speed for the onset of bubble-mediated CO₂ exchange. Therefore,
 formulations R3, R6, and R7 are applied only when U_{10} is higher than 5 m s⁻¹, and for lower
 wind speeds, Δ_s is set to zero.

• **Choice of Δ_s parameterisations.** Given that all three parameterisations yield consistent Δ_s
 estimates at high wind speeds, any of them can be used to reproduce the mechanistic results
 shown in Fig. 1. For simplicity, Equation R3 is used for the small-scale mechanism analysis.
 For large-scale impact assessment, Equations R3, R5, and R6 yield global ocean CO₂ flux
 increases by 0.41 Pg C yr⁻¹, 0.39 Pg C yr⁻¹, and 0.33 Pg C yr⁻¹, respectively. These estimates
 are lower than the previous estimate based on the original linear U_{10} -based Δ_s function.

- • **Response to “the scheme may mis-attribute part of the interfacial flux to the bubble**
**component”**. As previously discussed, Δ_s represents the supersaturation induced by bubbles,
that is partially offset by interfacial transfer processes. Therefore, it is not possible for Δ_s to
encompass the contribution from the interfacial component; rather, the interfacial transfer acts
to reduce the bubble-induced supersaturation.

Below is our revision of the manuscript based on the responses above.

Lines 150-170 (Section: Results-Evidence of asymmetric CO₂ transfer): Here, we use two
approaches to estimate Δ_s : reanalysis of the EC CO₂ data and derivation from existing gas transfer
velocity parameterisations. The detailed procedures for determining and parameterising Δ_s using
both methods are described in Materials and Methods, and here, we provide only a brief overview.
Both approaches require prior knowledge of δ and K_{int} . This study adopts the recent estimate of δ
for CO₂ from a bubble dynamic model ($\delta = 0.0132$; Leighton et al., 2018), and employs the K_{int}
parameterisation based on the EC DMS (dimethylsulfide) observations (Blomquist et al., 2017).
In the first method, we re-analyse the EC datasets to estimate Δ_s , which is then fitted against wind
speed (Fig. S5). This yields the following parameterisation:

$$339 \quad \Delta_s = 0.0132(1 - 1.37/U_{10}^{-0.37}), U_{10} \geq 5 \text{ m s}^{-1} \quad (5)$$

The alternative way to determine Δ_s is by linking Δ_s with the fractional contribution of bubble-
mediated gas transfer velocity to the total K (see Equation 4). If the widely-used ¹⁴C-based
parameterisation (Wanninkhof, 2014) is adopted to represent the total K , Δ_s can be derived as:

$$343 \quad \Delta_s = 0.0132(1 - 2.95/U_{10}^{-0.67}), U_{10} \geq 5 \text{ m s}^{-1} \quad (6)$$

For wind speeds below 5 m s⁻¹, Δ_s is set to zero for both parameterisations because bubble
contributions are negligible under this condition. The Δ_s values from Equations 5 and 6 diverge at
wind speeds below 10 m s⁻¹, but they converge at high wind speeds (10-20 m s⁻¹), with differences
of less than 10%. Both parameterisations yield comparable results for the subsequent analysis
within this section; therefore, only results based on Equation 5 are presented in the figures below.

Lines 230-257 (Section: Results-Impact of asymmetry on large-scale CO₂ flux estimates): Both Δ_s
parameterisations are used for this global ocean assessment. The global mean value of Δ_s is
estimated to be 0.004 (i.e., 0.4%) using Equation 5 and 0.003 (0.3%) using Equation 6.

The global ocean CO₂ uptake computed using the asymmetric equation is 0.33-0.41 Pg C yr⁻¹
greater than using the symmetric equation on average from 1991 to 2020, corresponding to ~15%
increase in the oceanic CO₂ sink estimates. Equation 5 produces higher Δ_s under typical oceanic
wind conditions (5-10 m s⁻¹; Fig. S5) and thus yields a larger $\Delta Flux$ magnitude (0.41 Pg C yr⁻¹)
than Equation 6 (0.33 Pg C yr⁻¹). This difference highlights the uncertainty associated with
quantifying Δ_s . The impact of the asymmetry on sea-air CO₂ flux is ubiquitous, but is most evident
in the Southern Ocean (South of 35°S) and relatively minor in the tropics (Fig. 2A). The Southern
Ocean accounts for about half of the asymmetry-induced flux increase in the global ocean. The
spatial variability of $\Delta Flux$ is primarily driven by wind speed (Fig. 2B), as stronger winds enhance
wave breaking and bubble formation, thereby amplifying asymmetric bubble-mediated transfer.
Notably, the impact of $\Delta Flux$ is always negative (i.e., enhanced ocean CO₂ uptake) because the
bubble over-pressure always favours gas invasion. Over the past three decades, $\Delta Flux$ has shown
a strengthening trend in a rate of ~3 Tg C yr⁻¹ per decade (Fig. 2C). This trend is primarily driven
by the rising atmospheric CO₂ concentration. In addition, hemispheric $\Delta Flux$ varies seasonally,
with greater asymmetrical fluxes in winter and smaller fluxes in summer (Fig. 2D). The seasonal
variability is primarily driven by seasonal wind variation and sea surface temperature changes.

**Fig. 2. Impact of asymmetric transfer on the sea-air CO₂ flux estimate ($\Delta Flux$).** (A) Map of $\Delta Flux$; (B)
 1°-latitude mean of $\Delta Flux$ and ERA5 wind speed; (C) Temporal trend in annual mean $\Delta Flux$; (D) Seasonal
 variations of $\Delta Flux$ in northern (green) and southern (purple) hemispheres (1-12 corresponding to January-
 December). The results shown here represent the ensemble mean $\Delta Flux$ estimated from two different Δ_s
 parameterisations (Equations 5 and 6). The $\Delta Flux$ shown in A, B, and D is averaged from 1991 to 2020. A
 negative $\Delta Flux$ means enhanced ocean CO₂ uptake.

**Line 338-342 (Discussion and conclusions):** Δ_s estimates from two different approaches are similar
 under high wind speeds ($U_{10} > 10 \text{ m s}^{-1}$), but differ substantially at lower wind speeds. This
 difference results in large variations in the estimated impact of bubble-induced asymmetry on
 global ocean CO₂ uptake, highlighting the need to reduce uncertainties in the Δ_s estimates.
 Nevertheless, the value of Δ_s (0.3-0.4% on average) estimated in this study is consistent with
 existing evidence.

**Lines 397-427 (Materials and Methods-Estimation of the asymmetry factor Δ_s):** The asymmetry
 factor (Δ_s) in Equation 2 is a key parameter in this study. We estimate Δ_s using two approaches.

Both methods rely on the independent estimates of the over-pressure factor (δ), the interfacial
transfer velocity (K_{int}) and the total gas transfer velocity (K) (see Equation 4). For CO_2 , δ is
primarily driven by the hydrostatic pressure and is directly related to the effective penetration
depth of the bubble plume, which has been shown to remain largely unchanged with wind speed
(Keeling, 1993; Czerski et al., 2022). Accordingly, we adopt a fixed δ value of 0.0132, simulated
from a bubble dynamic model based on near-surface bubble observations (Leighton et al., 2018).
For K_{int} , we use transfer velocity parameterisations based on EC DMS observations (Blomquist et
al., 2017; Fig. S8), as the high solubility of DMS minimises the contribution of bubbles to its
exchange. Based on the way to represent K , two approaches are proposed to estimate Δ_s .

In the first approach, hourly EC sea-air CO_2 flux measurements are treated as the total CO_2
exchange in the estimate of estimate Δ_s . However, these EC CO_2 fluxes inherently include the
effect of chemical enhancement (CE), which is absent in EC DMS observations that are being used
for K_{int} . While CE is negligible at high wind speeds, it becomes relatively important under low
wind conditions. Given that Δ_s is sensitive to the treatment of K at low wind speeds, CE
contribution should be removed from the observed EC CO_2 transfer velocities. To account for this,
we applied a CE correction ratio derived from analytical and numerical models (Luhar et al., 2018;
Fairall et al., 2022). We then combine Equations 2 and 4 to derive Δ_s for each corresponding flux
using an iterative method (see Supplementary Material, Section 3). The resulting Δ_s increases with
the wind speed and asymptotically approaches the value of δ when the bubble-mediated exchange
dominates the total gas exchange (Fig. S5). Given that $\Delta_s = \delta K_{bub} / K = \delta(K - K_{int}) / K$, and K and
K_{int} can be expressed as $a_1 U_{10}^{b_1}$ and $a_2 U_{10}^{b_2}$, respectively, Δ_s can thus be parameterised as $\Delta_s = \delta(1$
$- a_3 / U_{10}^{b_3})$. Fitting the bin averages of Δ_s for U_{10} between 5-20 m s^{-1} yields Equation 5.

Alternatively, the widely used ^{14}C inventory-based parameterisation (Wanninkhof, 2014) can
represent the total K , and the difference between K and K_{int} corresponds to the K_{bub} (Fig. S5A).
Thus, Δ_s can be directly calculated using the existing parameterisations of K and K_{int} , which yields
Equation 6. If the recent K parameterisation based on the synthesis of the EC sea-air CO_2 data
(Yang et al., 2022) is used to represent the total gas transfer velocity, the derived Δ_s will be similar
to Equation 5 (Fig. S5B), and thus is not shown in the main text.

We also slightly adjust some texts the Abstract and Discussions accordingly. The figures and Table
S1 in the supplementary material are also updated using the new Δ_s parameterisation (Equation 5),

but there are no significant changes. Please refer to the manuscript with tracks of the change for
relevant minor adjustments.

**References**

Blomquist, B. W., Brumer, S. E., Fairall, C. W., Huebert, B. J., Zappa, C. J., Brooks, I. M., et al.
(2017). Wind Speed and Sea State Dependencies of Air-Sea Gas Transfer: Results From the
High Wind Speed Gas Exchange Study (HiWinGS). *Journal of Geophysical Research:*
*Oceans*, 122(10), 8034–8062. <https://doi.org/10.1002/2017JC013181>

Czerski, H., Brooks, I. M., Gunn, S., Pascal, R., Matei, A., & Blomquist, B. (2022). Ocean
bubbles under high wind conditions—Part 1: Bubble distribution and development. *Ocean*
*Science*, 18(3), 565–586.

Dong, Y., Jähne, B., Woolf, D. K., Krall, K. E., Yang, M., Czerski, H., et al. (2025). The Role of
Bubbles in Air-Sea Gas Exchange: A Critical Review. *Authorea Preprints*.

Fairall, C. W., Yang, M., Brumer, S. E., Blomquist, B. W., Edson, J. B., Zappa, C. J., et al.
(2022). Air-Sea Trace Gas Fluxes: Direct and Indirect Measurements. *Frontiers in Marine*
*Science*, 9(July), 1–16. <https://doi.org/10.3389/fmars.2022.826606>

Keeling, R. F. (1993). On the role of large bubbles in air-sea gas exchange and supersaturation in
the ocean. *Journal of Marine Research*, 51(2), 237–271.
<https://doi.org/10.1357/0022240933223800>

Leighton, T. G., Coles, D. G. H., Srokosz, M., White, P. R., & Woolf, D. K. (2018). Asymmetric
transfer of CO₂ across a broken sea surface. *Scientific Reports*, 8(1), 1–9.
<https://doi.org/10.1038/s41598-018-25818-6>

Luhar, A. K., Woodhouse, M. T., & Galbally, I. E. (2018). A revised global ozone dry deposition
estimate based on a new two-layer parameterisation for air–sea exchange and the multi-year
MACC composition reanalysis. *Atmos. Chem. Phys.*, 18(6), 4329–4348.
<https://doi.org/10.5194/acp-18-4329-2018>

Wanninkhof, R. (2014). Relationship between wind speed and gas exchange over the ocean
revisited. *Limnology and Oceanography: Methods*, 12(JUN), 351–362.

<https://doi.org/10.4319/lom.2014.12.351>

Wanninkhof, R., & Knox, M. (1996). Chemical enhancement of CO₂ exchange in natural waters.

*Limnology and Oceanography*, 41(4), 689–697.

Wanninkhof, R., Asher, W. E., Ho, D. T., Sweeney, C., & McGillis, W. R. (2009). Advances in

Quantifying Air-Sea Gas Exchange and Environmental Forcing. *Annual Review of Marine*

*Science*, 1(1), 213–244. <https://doi.org/10.1146/annurev.marine.010908.163742>

We thank both reviewers for their positive feedback and especially Reviewer #3 for identifying
two important typos in our manuscript.

**Reviewer #3 (Remarks to the Author):**

The authors have responded my remarks in an appropriate manner. I found their response to
Reviewer 4 concerns are convincing. This led to an improvement of the article by reinforcing the
physical soundness of the analytical approach and the statistical robustness of the results. The
manuscript is clear and well written.

Thank you very much for your effort to help us improve the manuscript.

Here are some typos that should be corrected:

- Equations (5) and (6) (and R6 and R7) should read

$\Delta_s = 0.0132(1 - 1.37U_{10}^{-0.37})$

$\Delta_s = 0.0132(1 - 2.95U_{10}^{-0.67})$

as in Figure S5 caption.

Thank you for pointing out this typo, which is very important. We have revised them:

$$\Delta_s = 0.0132(1 - 1.37U_{10}^{-0.37}), U_{10} \geq 5 \text{ m s}^{-1} \quad (5)$$

$$\Delta_s = 0.0132(1 - 2.95U_{10}^{-0.67}), U_{10} \geq 5 \text{ m s}^{-1} \quad (6)$$

- Line 425 « represent »

Thank you. Revised: ... (Yang et al., 2022) is used to **represent** the total gas transfer velocity...

**Reviewer #4 (Remarks to the Author):**

The authors have presented a clear and thorough explanation of their methodology and provided
comprehensive responses demonstrating the effectiveness of their approach. I find the revisions
satisfactory and the responses well-answered. The manuscript has been significantly improved in
clarity, structure, and scientific rigor. I believe the revised version meets the standards for
publication.

Thank you for the encouraging feedback. We greatly appreciate your valuable comments on Δ_s .